# METAFLOW: A META APPROACH OF TRAINING LLMS INTO GENERALIZABLE WORKFLOW GENERATORS

## ABSTRACT

Large language models (LLMs) excel across a wide range of tasks, yet their instance-specific solutions often lack the structural consistency needed for reliable deployment. Workflows that encode recurring algorithmic patterns at the task level provide a principled framework, offering robustness across instance variations, interpretable traces for debugging, and reusability across problem instances. However, manually designing such workflows requires significant expertise and effort, limiting their broader application. While automatic workflow generation could address this bottleneck, existing methods either produce instance-specific solutions without learning task-level patterns, or cannot generalize beyond their training configurations. We present **MetaFlow**, which casts workflow generation as a meta-learning problem: given a task and an operator set, the model learns to compose solution strategies. MetaFlow trains in two stages—supervised fine-tuning on synthetic workflow data, followed by reinforcement learning with verifiable rewards (RLVR) that uses execution feedback across problem instances in the task to improve end-to-end success. The resulting model produces effective workflows for trained tasks and exhibits strong generalization to untrained tasks and novel operator sets. Across benchmarks in question answering, code generation, and mathematical reasoning, MetaFlow achieves performance comparable to state-of-the-art baselines on in-domain tasks with single inference, while demonstrating remarkable zero-shot generalization capabilities on out-of-domain tasks and operator sets.

## 1 INTRODUCTION

Large Language Models (LLMs) have demonstrated significant performance across a wide range of tasks, including code generation, question answering, and mathematical reasoning (Austin et al., 2021; Chen et al., 2021; Yang et al., 2018; Dua et al., 2019; Ding et al., 2024; Jiang et al., 2025; Cobbe et al., 2021; OpenAI, 2023; Zhu et al., 2024). However, because these models generate instance-specific solutions, they lack the structural consistency and transparency needed for reliable deployment, while also being difficult to adapt to similar tasks. *Workflows* that encode recurring algorithmic patterns provide a principled alternative, decomposing complex challenges into structured, manageable steps. However, manually designing such workflows requires significant expertise and effort, limiting their broader application.

To address this challenge, recent effort have focused on the automatic workflow generation (Khattab et al., 2023; Li et al., 2024; Song et al., 2024; Zhang et al., 2024a). Endowing LLMs with this strategy planning capability means lowering the barrier for complex task automation from requiring manual programming by experts to merely providing high-level task descriptions, thereby greatly liberating productivity. Nevertheless, representing the workflow as static graph (Zhuge et al., 2024) or neural network (Liu et al., 2024) in many of these methods limits the flexibility of generatable workflows.

A promising direction emerges from works like ADAS (Hu et al., 2024), AFlow (Zhang et al., 2024b), ScoreFlow (Wang et al., 2025) and FlowReasoner (Gao et al., 2025), which represent workflows as code (a structured combination of predefined *operators*), making the automatic generation of workflows more flexible and expressive, where *operator* is an encapsulation of common agentic

operations introduced by Zhang et al. (2024b). Within this code-based framework, current approaches adopt two different paradigms for workflow generation.

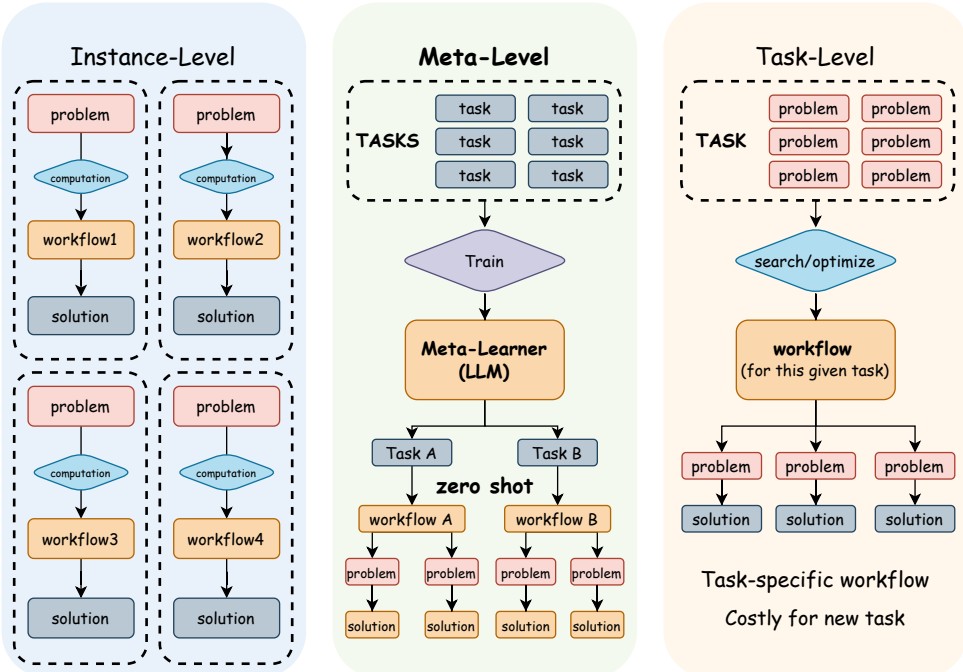

Figure 1: Illustration of **instance-level**, **task-level**, and **meta-level (ours)** workflow generation approaches. Unlike **instance-leve** methods that generate workflows for individual problems or **task-level** methods that require costly search for each new task, our meta-learning approach enables zero-shot workflow generation across tasks.

The first paradigm comprises **task-level** approaches, exemplified by ADAS (Hu et al., 2024) and AFlow (Zhang et al., 2024b), which formulate workflow generation as a search problem within predefined task-operator set combination. Both methods employ iterative search strategies, with ADAS using evolutionary algorithms and AFlow using Monte Carlo Tree Search (MCTS), to discover high-performing workflows through repeated refinement. However, this search-based paradigm inherently constrains them to specific task and predetermined operator set. When encountering new tasks or operators, these methods require complete re-optimization from scratch, incurring substantial computational costs (Wang et al., 2025).

Conversely, the second paradigm consists of **instance-level** approaches, exemplified by Score-Flow (Wang et al., 2025) and FlowReasoner (Gao et al., 2025), which generate workflows tailored to individual problem instances in the task. Both methods dynamically construct workflows at inference time, with ScoreFlow leveraging gradient-based optimization to refine agentic workflows, and FlowReasoner employing reasoning chains distilled from advanced models to design query-specific multi-agent systems. While these instance-level methods excel at tailoring workflows to specific problem instances, this granularity comes at the cost of reusability and deployment efficiency. They cannot capture task-level patterns that recur across similar problem instances, leading to redundant workflow generation for each query. In deployment scenarios, this approach foregoes the benefits of having optimized, reusable workflow templates that could be consistently applied to entire task.

The limitations of existing paradigms underscore two fundamental challenges that must be addressed to achieve truly general-purpose automatic workflow generation. **(1) How can we overcome the re-optimization requirement of task-level approaches when facing new domains (task-operator set combinations)? (2) How can we learn generalizable patterns that avoid redundant instance-level generation while adapting effectively to untrained domains?**

To systematically overcome the challenges, we propose **MetaFlow**, which formulates workflow generation as a meta-learning problem. As illustrated in Figure 1, unlike task-level search-based meth-

ods that require expensive re-optimization for new task-operator set combinations, and instance-level methods that generate workflows for individual queries, **MetaFlow** learns to directly synthesize workflows from task descriptions and operator set specifications, enabling zero-shot generation through a single model inference.

To achieve robust zero-shot generalization, **MetaFlow** employs a two-stage training paradigm with diverse task-operator pairs. Adopting the code-based workflow representation from prior works(Hu et al., 2024; Zhang et al., 2024b; Wang et al., 2025; Gao et al., 2025), we first synthesize thousands of workflows using Qwen-Max (Team, 2024) across four tasks and a single operator set to finetune Qwen3-8B (Yang et al., 2025), establishing the foundation for understanding how tasks and operators relate to workflow structures. Subsequently, we apply online reinforcement learning with GRPO (Shao et al., 2024) across expanded domains, where execution feedback on problem instances directly optimizes the generation policy. This training ensures the model learns generalizable workflow construction principles rather than memorizing patterns. At inference, **MetaFlow** zero-shot generates effective workflows for novel configurations with only a single forward pass.

Our main contributions are:

- **Meta-learning Framework**: We introduce **MetaFlow**, a novel approach that reformulates workflow generation from discrete search within fixed configurations to continuous learning across diverse task-operator set combinations. By conditioning workflow generation on task descriptions and operator specifications, our framework achieves strong zero-shot generalization to unseen domains without any re-optimization, reducing computational cost from thousands of API calls to a single model inference.

- **Scalable Training Pipeline**: We design a two-stage training framework combining supervised learning with online reinforcement learning, utilizing diverse domains to ensure robust generalization to untrained domains.

- **Comprehensive Evaluation**: Extensive experiments demonstrate that **MetaFlow** achieves competitive performance on in-domain benchmarks while exhibiting remarkable zero-shot generalization to out-of-domain task classes and operator sets, including solving programming problems with novel operator combinations never seen during training and solving question answering problem using the vector database search operator.

## 2 RELATED WORKS

### 2.1 AGENTIC WORKFLOW

Agentic workflows decompose complex tasks into structured steps through predefined operators and dependencies (Zhang et al., 2024b; Wang et al., 2025; Gao et al., 2025). Unlike autonomous agents that learn through environment interaction (Zhuge et al., 2024; Hong et al., 2024), workflows provide interpretable and consistent execution patterns. Recent works adopt code-based representations for superior expressiveness (Hu et al., 2024; Zhang et al., 2024b; Wang et al., 2025; Gao et al., 2025), supporting applications in code generation, question answering, and mathematical reasoning (Austin et al., 2021; Chen et al., 2021; Yang et al., 2018; Dua et al., 2019; Ding et al., 2024; Jiang et al., 2025; Cobbe et al., 2021; OpenAI, 2023; Zhu et al., 2024). However, manual workflow design remains a significant bottleneck requiring deep expertise.

### 2.2 AUTOMATIC WORKFLOW GENERATION

Recent advances have explored automating workflow generation for improving LLM performance (Chen et al., 2023; Zhang et al., 2024b; Wang et al., 2025; Li et al., 2024; Song et al., 2024). While some methods optimize prompts within fixed workflows (Guo et al., 2023; Khattab et al., 2023), we focus on optimizing workflow structures directly.

Current structural optimization follows two paradigms. Task-level approaches like ADAS (Hu et al., 2024) and AFlow (Zhang et al., 2024b) search for optimal workflows through evolutionary algorithms or MCTS, but require complete re-optimization for new domains. Instance-level methods including ScoreFlow (Wang et al., 2025) and FlowReasoner (Gao et al., 2025) generate query-specific workflows but fail to extract reusable patterns.

Our Methods, **MetaFlow** reformulates workflow generation as meta-learning over diverse task-operator combinations during training. Through two-stage optimization combining supervised learning with reinforcement learning, it achieves true zero-shot generation—producing effective workflows for novel domains via single model inference, eliminating both re-optimization and adaptation overhead.

## 3 PROBLEM DEFINITION

Existing works often formulate automatic workflow generation as optimization problems (Xu et al., 2025; Li et al., 2025), requiring separate optimizations for each task. We elevate this perspective by reformulating it as a meta-learning problem (Finn et al., 2017; Franceschi et al., 2018). To ground this formulation, we first define our core concepts:

- PROBLEM INSTANCE p: A single, concrete problem to be solved.
- TASK C: A family of problem instances sharing a common structure and solution strategy (e.g., GSM8K mathematical reasoning, DROP reading comprehension).
- OPERATOR SET Ops: A collection of fundamental, reusable operations (Zhang et al., 2024b) (e.g., Generate, Revise, Ensemble).
- DOMAIN $(C, Ops)$: The combination of a task and an operator set, defining a complete problem-solving context.

Within this meta-learning framework, an LLM serves as the meta-learner (the planner $\pi_\theta$). Its core responsibility is to learn a meta-strategy that enables fast adaptation: given any domain $(C, Ops)$, the planner rapidly generates an efficient and reusable WORKFLOW W—a structured sequence of operators from Ops. When executed on any PROBLEM INSTANCE p from task C, this workflow produces a high-quality SOLUTION s.

Unlike traditional meta-learning approaches such as Model-Agnostic Meta-Learning (MAML) (Finn et al., 2017), which rely on gradient updates for adaptation, our method achieves fast adaptation through the synthesis of workflows without requiring any gradient-based fine-tuning.

As a meta-learning task, our goal is to optimize the meta-parameters $\theta$ of the planner $\pi_\theta$ such that the generated workflows maximize expected rewards across the domain distribution $\mathcal{D}_{(C,Ops)}$. We adopt the classic **bi-level optimization** framework Franceschi et al. (2018):

$$\theta^* = \arg\max_\theta \underbrace{\mathbb{E}_{(C,Ops)\sim\mathcal{D}_{(C,Ops)}}}_{\text{Outer Loop: Meta-Optimization}} \left[ \underbrace{\mathbb{E}_{W\sim\pi_\theta(\cdot|C,Ops)}[\mathbb{E}_{p\sim C}[R(\text{Exec}(W, p))]]}_{\text{Inner Loop: Fast Adaptation \& Evaluation}} \right]$$

where:

- $(C, Ops) \sim \mathcal{D}_{(C,Ops)}$: Sample a domain (task-operator pair) from the distribution.
- $W \sim \pi_\theta(\cdot \mid C, Ops)$: The planner performs *fast adaptation*—given task description C and operator specifications Ops, it generates a customized workflow W without gradient updates.
- $p \sim C$: Sample problem instances from task C to evaluate the workflow.
- $\text{Exec}(W, p)$: Execute workflow W on problem p to produce a solution.
- $R(\cdot)$: Reward function evaluating solution quality (e.g., test pass rate, answer correctness).

The **outer loop** optimizes performance across the entire domain distribution $\mathcal{D}_{(C,Ops)}$ rather than on individual tasks or instances, driving the meta-learner $\pi_\theta$ to acquire cross-domain generalization capabilities. This contrasts sharply with prior works:

- **Instance-level methods** (e.g., ScoreFlow (Wang et al., 2025), ComfyUI-R1 (Xu et al., 2025)) optimize workflows for individual problem instances: $\arg\max_{W_p} \mathbb{E}_{p\sim C}[R(\text{Exec}(W_p, p))]$.
- **Task-level methods** (e.g., AFlow (Zhang et al., 2024b)) optimize a single workflow per task but require separate optimization for each new task-operator combination.

Our framework elevates to the **meta-level**, learning a planner that generalizes across tasks and operators: $W \sim \pi_\theta(\cdot \mid C, Ops)$.

# 4 METHODOLOGY

Our **MetaFlow** framework aims to train a large language model through a two-stage learning process to automatically generate efficient and reusable workflows for given domains $(C, Ops)$, where $C$ represents a task and $Ops$ denotes the available operator set. The framework addresses a key challenge in automated workflow generation: how to enable a model to quickly adapt to new domains (task-operator combinations) without requiring extensive training data for each. This section presents our system architecture (Section 4.1) and the two-stage training algorithm combining supervised fine-tuning and reinforcement learning with verifiable rewards (Section 4.2).

## 4.1 METAFLOW ARCHITECTURE

The core architecture of **MetaFlow** consists of two components: (1) a **planner LLM** $\pi_\theta$ that generates workflows conditioned on domain specifications $(C, Ops)$, and (2) an **execution-evaluation environment** that orchestrates workflow execution using the MetaGPT framework (Hong et al., 2023), invokes operators on sampled problem instances, and computes verifiable reward scores based on correctness evaluation.

**Workflow Representation**: A workflow is represented as a structured script based on the MetaGPT framework (Hong et al., 2023), composed of a series of predefined operator calls. As illustrated in the left part of Figure 2, we adopt the foundational text-processing operators from AFlow (Zhang et al., 2024b), including `Generate`, `Summarize`, `Revise`, `Ensemble`, and `Programmer` for basic tool invocation. Building upon the same MetaGPT interface, we further introduce additional operators for advanced text processing (e.g., `Decompose 1`, `SelfConsistency`) and domain-specific tools (e.g., `VectorSearch 3`). This standardized operator interface serves two purposes: (1) supervised fine-tuning provides a cold start that ensures the model generates syntactically valid operator calls with higher probability, and (2) the introduction of new operators demonstrates the necessity of our meta-learning approach for efficient and transferable workflow generation across diverse operator sets. Implementation details of the operators are provided in Appendix A.1 for reference.

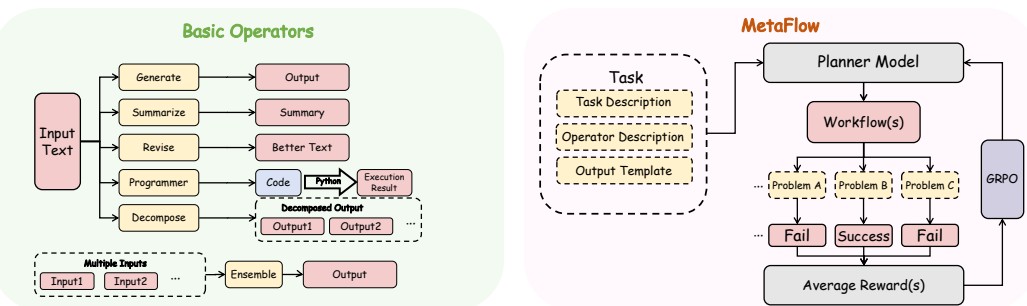

Figure 2: Overview of MetaFlow architecture. **Left:** Basic operators and their functionalities, including text-processing operators (`Generate`, `Summarize`, `Revise`, `Ensemble`) and basic tool-invocation operators (`Programmer`). **Right:** The MetaFlow training framework showing how the planner model generates workflows conditioned on domain $(C, Ops)$, which are then evaluated on problem instances to compute rewards for GRPO optimization.

**Input-Output**: As illustrated in the right part of Figure 2, the MODEL INPUT consists of two parts: TASK DESCRIPTION 4, which elucidates the characteristics of the target task $C$ and the input-output formats of problem instances; OPERATOR DESCRIPTIONS 5, which define the functions, parameters, and input-output formats of each available operator in $Ops$ (whether pre-defined or user-defined). The OUTPUT 6 is a structured WORKFLOW aimed at efficiently solving all problem instances from task $C$ using operator set $Ops$. This design paradigm allows users to introduce new operators and tasks through natural language descriptions at inference time, enabling the model to rapidly adapt without additional training. Implementation details are provided in Appendix A.2.

**Execution & Evaluation Environment**: To achieve end-to-end optimization, we build an automated environment. This environment receives a candidate workflow $\mathsf{W}_i$ and a set of $N$ problem instances $\mathsf{p}_1, \ldots, \mathsf{p}_N$ sampled from task $\mathsf{C}$ as input. In the execution phase, the environment uses the MetaGPT framework to orchestrate the workflow (Hong et al., 2023). Each operator in the workflow (such as `Generate`) completes its specific subtask by calling an external lightweight language model API (e.g., `Qwen-Turbo` or `GPT-4o-mini-0718`) and processes each problem instance. Upon completion, an automated evaluator module verifies the correctness of each execution result, for example, by running unit tests or comparing outputs with ground truth answers. Based on the evaluation results, the environment computes a verifiable reward score $R(\mathsf{W}_i)$ for workflow $\mathsf{W}_i$, calculated as the average success rate over $N$ instances:

$$R(\mathsf{W}_i) = \frac{1}{N} \sum_{j=1}^{N} \mathbb{I}(\text{is correct}(\text{Exec}(\mathsf{W}_i, \mathsf{p}_j))) \in [0, 1] \tag{1}$$

where $\mathbb{I}(\cdot)$ is the indicator function. This reward score $R(\mathsf{W}_i)$ is then passed to the training algorithm as the basis for policy updates.

## 4.2 TRAINING ALGORITHM

The training process consists of two stages: **supervised fine-tuning (SFT)** and **Reinforcement Learning with Verifiable Reward (RLVR)**. This design leverages supervised data to establish syntactic correctness, then employs reinforcement learning to optimize workflow effectiveness, separating structural learning from performance optimization.

### 4.2.1 PHASE ONE: SFT INITIALIZATION

To address the cold start problem in reinforcement learning, we first perform **supervised fine-tuning (SFT)** on the base LLM using a dataset of $([\mathsf{C}, \mathsf{Ops}], \mathsf{W})$ pairs, which is the pairs of domain $[\mathsf{C}, \mathsf{Ops}]$ and the corresponding workflow $\mathsf{W}$. This phase teaches the model to generate syntactically correct workflows following the required template structure (Listing 6) and proper operator invocation patterns (Listing 5). By ensuring the model $\pi_\theta$ can produce well-formed code, the subsequent RL phase can focus solely on optimizing workflow effectiveness rather than learning basic syntax, significantly narrowing the search space and accelerating convergence. Details of the SFT dataset construction, including the four-stage pipeline and LoRA configuration, are provided in Appendix B.

### 4.2.2 PHASE TWO: RLVR OPTIMIZATION

After SFT, we employ policy gradient algorithms to perform end-to-end self-improvement on the planner $\pi_\theta$. The core of this phase is the **Reinforcement Learning with Verifiable Reward (RLVR)** loop, with the complete algorithm presented in Algorithm 1. The specific process consists of the following steps:

1. **Policy Sampling**: For a task $\mathsf{C}$ sampled from the training set, the planner $\pi_\theta$ generates a batch of $k$ candidate workflows $\mathsf{W}_1, \mathsf{W}_2, \ldots, \mathsf{W}_k$.

2. **Execution & Reward Calculation**: Each candidate workflow $\mathsf{W}_i$ is tested in the execution and evaluation environment described in Section 4.1, obtaining its corresponding reward score $R(\mathsf{W}_i)$ (Equation 1) that reflects generalization capability:

$$C \in \mathcal{D}_{\text{train}} \stackrel{\pi_\theta}{\Longrightarrow} \{W_i | 1 \le i \le k\} \stackrel{\text{each } W_i}{\Longrightarrow} \{[W_i, R(W_i)], \text{ executed on } p_1^{(i)}, \cdots, p_N^{(i)} | 1 \le i \le k\}$$

3. **Policy Update**: We use this batch of $[W_i, R(W_i)]$ pairs to update the parameters of the planner $\pi_\theta$ using the **Group Relative Policy Optimization (GRPO)** algorithm (Shao et al., 2024). The core idea of GRPO is to use the average performance within the group as a baseline to estimate advantages, thereby avoiding training an independent value network.

$$\text{advantage} = \hat{A}(\mathsf{W}_i) = R(\mathsf{W}_i) - \mu_R, \ \mu_R = \sum_{j=1}^{k} R(\mathsf{W}_j)/k$$

This group-relative advantages makes the optimization signal derive from whether the workflow performance is better or worse than the current batch's average level, rather than an absolute, potentially noisy value estimate.

4. **Variance Reduction: Evaluation Based on Common Random Numbers**: The effectiveness of policy gradients largely depends on the accuracy of the advantage function $\hat{A}(\mathsf{W}_i)$ estimation. In our GRPO method, the advantage is computed relative to the batch average reward $\mu_R$. If each workflow $\mathsf{W}_i$ in the batch is evaluated on a set of independently and randomly sampled problem instances, the variance of the reward $R(\mathsf{W}_i)$ will include not only policy randomness but also environmental randomness. This additional variance propagates to $\mu_R$ and $\hat{A}(\mathsf{W}_i)$, producing noisier gradients and reducing learning efficiency.

To address this issue, we adopt a classic variance reduction technique: **Common Random Numbers (CRN)** (Kleijnen, 1975). In specific implementation, we ensure that all $k$ candidate workflows $\mathsf{W}_1, \ldots, \mathsf{W}_k$ in the same training batch are evaluated on the exact same set of problem instances $\{\mathsf{p}_1, \ldots, \mathsf{p}_N\}$. By fixing the random variable of problem instances when comparing workflows, we eliminate noise arising from differences in problem sampling. This keeps the expected value of $R(\mathsf{W}_i) - R(\mathsf{W}_j)$ unchanged but significantly reduces its variance. Ultimately, this ensures that our computed relative advantages more accurately reflect the intrinsic performance differences between workflows, leading to a more stable policy update direction and accelerating model convergence.

---

**Algorithm 1** GRPO Optimization with CRN

---

**Require:** Training tasks $\mathcal{D}_{\text{train}}$, initial policy $\pi_\theta$, group size $k$, batch size $B$, iterations $T$
1: **for** $t = 1, \ldots, T$ **do**
2:     **Batch Sampling:** Sample tasks $\{\mathsf{C}_1, \ldots, \mathsf{C}_B\} \sim \mathcal{D}_{\text{train}}$
3:     **for** each $\mathsf{C}_b$ in batch **do**
4:         **Policy Sampling:** Generate $k$ workflows $\mathcal{W}_b = \{\mathsf{W}_{b,1}, \ldots, \mathsf{W}_{b,k}\}$
5:           where $\mathsf{W}_{b,i} \sim \pi_\theta(\cdot | \mathsf{C}_b)$
6:         **Common Random Numbers:** Fix problem set $\mathcal{P}_b = \{\mathsf{p}_1, \ldots, \mathsf{p}_N\} \sim \mathsf{C}_b$
7:         **Reward Computation:** For all $i \in [1, k]$:
8:           $R(\mathsf{W}_{b,i}) = \frac{1}{N} \sum_{j=1}^{N} \mathbb{I}[\mathsf{W}_{b,i} \text{ solves } \mathsf{p}_j]$
9:         **Group-Relative Advantage:** Compute baseline $\mu_{R,b} = \frac{1}{k} \sum_{i=1}^{k} R(\mathsf{W}_{b,i})$
10:          For all $i$: $\hat{A}(\mathsf{W}_{b,i}) = R(\mathsf{W}_{b,i}) - \mu_{R,b}$
11:     **end for**
12:     **Batch GRPO Update:**
13:         Collect all $\{(\mathsf{W}_{b,i}, \hat{A}(\mathsf{W}_{b,i}))\}_{b=1,\ldots,B; i=1,\ldots,k}$
14:         Update: $\theta \leftarrow \text{GRPO}(\theta, \{(\mathsf{W}_{b,i}, \hat{A}(\mathsf{W}_{b,i}))\})$
15: **end for**
16: **return** Optimized policy $\pi_\theta$

---

## 5 EXPERIMENTS

This section evaluates **MetaFlow** through systematic experiments addressing three core questions: (1) Can **MetaFlow** achieve competitive performance with single-inference generation, eliminating the computational overhead of per-instance optimization? (2) Does the framework generalize to out-of-distribution tasks and novel operators unseen during training? (3) How do design choices—task-level versus instance-level generation, simple versus complex workflows—affect the performance-cost trade-off? We present training configuration, operator integration experiments, zero-shot generalization results, and main performance analysis.

### 5.1 TRAINING CONFIGURATION

**Stage 1: Supervised Fine-Tuning (SFT) Initialization.** We first conduct supervised fine-tuning (SFT) on the base model $\mathtt{Qwen3\text{-}8B}$ to teach the model $\pi_\theta^{\text{Base}}$ to generate syntactically correct workflows (Listing 6) and narrow the search space. We construct an expert dataset containing approximately $1,300$ high-quality $([\mathsf{C}, \mathsf{Ops}], \mathsf{W})$ pairs, covering the tasks $\mathcal{D}_{\text{train}}^{\text{SFT}} = \{\text{GSM8K, DROP, MBPP, Humaneval}\}$ with the corresponding operator set for each task $\mathsf{Ops}^{\text{SFT}} =$

(Generate, Summarize, Revise, Ensemble). The dataset is synthesized using Qwen-Max API through a four-stage pipeline (Appendix B). As shown in Figure 3 (left and middle), we train for one epoch with batch size 16 using LoRA (rank-16) (Hu et al., 2022) to prevent mode collapse on this limited dataset (Appendix B.2).

**Stage 2: Reinforcement Learning with Verifiable Rewards (RLVR) Optimization.** Following SFT initialization, we employ RLVR for end-to-end optimization of the planner $\pi_\theta^{\text{SFT}}$ with the same task set $\mathcal{D}_{\text{train}}^{\text{RLVR}} = \mathcal{D}_{\text{train}}^{\text{SFT}} = \{\text{GSM8K, DROP, MBPP, Humaneval}\}$. Critically, to enhance generalization to diverse operator configurations, we train with four different operator sets $\{\text{Ops}^{\text{RLVR},i}\}_{i=1}^4$ that progressively introduce novel operators (Programmer, Decompose) beyond the base SFT set, randomly sampling domain combinations $(\text{C, Ops})$ at each iteration. The training employs Algorithm 1 for 137 steps. To control training costs, the execution of generated workflow $\text{W}_i$ on problem instance $\text{p}_j$ calls the Qwen-Turbo API. As shown in Figure 3 (right), the average reward rapidly increases from 0.67 to approximately 0.88 within the first 35 training steps, then stabilizes around 0.85–0.90, demonstrating the effectiveness of GRPO optimization with CRN. We select the checkpoint at step 100 for evaluation. The details are provided in Appendix C.

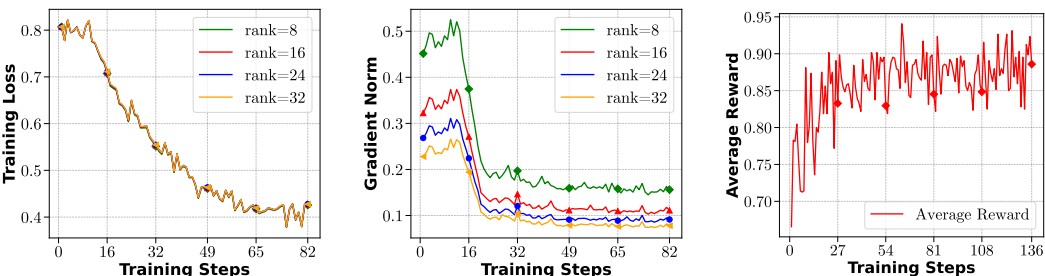

Figure 3: Training dynamics of MetaFlow. **Left:** SFT training loss across different LoRA ranks (with the same learning rate $lr = 0.00002$). **Middle:** SFT gradient norms showing convergent behavior. **Right:** RLVR average reward demonstrating rapid policy improvement and stabilization.

## 5.2 LOW-COST OPERATOR INTEGRATION: CONTINUOUS LIBRARY EXPANSION

A key advantage of **MetaFlow** over task-level search methods (Zhang et al., 2024b; Hu et al., 2024) is its ability to continuously expand the operator library without costly re-optimization. To validate this capability, we demonstrate a practical workflow: abstracting a manually-tested reasoning pattern into a reusable operator. Specifically, we introduce the SelfConsistency operator (see Appendix 2 for complete implementation), which encapsulates the Self-Consistency pattern (Wang et al., 2022) (parallel generation with majority voting), directly derived from one of our experimental baselines (CoT SC in Table 1). Critically, incorporating this operator requires no model retraining, only defining its natural language interface following the format described in Listing 5.

To demonstrate immediate operator utilization, we generate workflows for GSM8K mathematical reasoning after integrating SelfConsistency. Listing 13 shows an example generated workflow that successfully invokes the newly integrated operator with parallel sampling (n=5) and similarity-based voting, demonstrating that **MetaFlow** can immediately compose effective workflows with novel operators through single-inference generation without any retraining.

**Implications.** This validates **MetaFlow**'s *low-cost continuous extensibility*: (1) **Minimal integration cost**: introducing a new operator requires only defining its interface, without model retraining or iterative search (hours for MCTS/evolutionary methods vs. minutes for **MetaFlow**); (2) **Rapid abstraction of proven patterns**: practitioners can quickly encapsulate manually-tested workflow logic discovered through experimentation into reusable operators; (3) **Immediate high-quality generation**: generating effective workflows with the new operator costs merely a single inference; (4) **Scalable library growth**: each new operator becomes immediately available across all tasks without proportional computational overhead, enabling continuous knowledge accumulation. This paradigm transforms workflow optimization from isolated task-specific searches into a scalable knowledge base where proven patterns become reusable primitives.

## 5.3 Zero-Shot Generalization to Novel Operators

To validate true out-of-distribution (OOD) generalization, we evaluate **MetaFlow** on Hot-potQA (Yang et al., 2018) multi-hop question answering with the `VectorSearch` operator (Listing 3)—entirely unseen during training. This tests two critical OOD dimensions: (1) **domain shift** from math/code reasoning to retrieval-based QA, and (2) **novel operator** requiring hybrid document and sentence-level vector search. We generate 100 candidate workflows with Ops = {`Generate`, `Summarize`, `Revise`, `Ensemble`, `VectorSearch`} evaluated on 100 HotpotQA instances (Validation), compared against a CoT+RAG baseline using the same API (`Qwen-Turbo`).

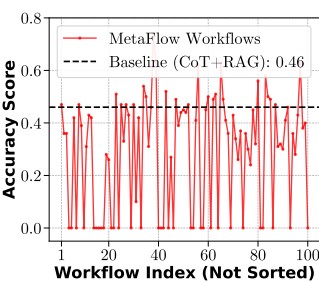 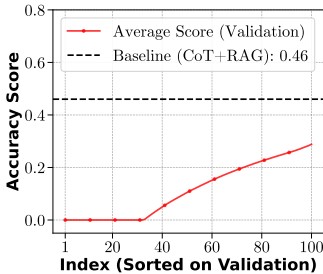 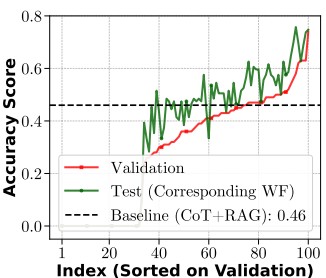

Figure 4: Zero-shot generalization with novel `VectorSearch` operator on HotpotQA. **Left:** Individual workflow performance across 100 generations shows high variance, 31 workflows fail completely while the best achieves 0.74 accuracy. **Middle:** Average score converges to 0.29, reflecting the exploration cost of zero-shot generation. **Right:** Best-of-$k$ score reaches 0.74 (60.9% above baseline 0.46), surpassing baseline at $k = 76$.

Figure 4 reveals the characteristics of zero-shot workflow generation. The **left panel** shows substantial performance variance: while 31% of workflows fail (score=0.0, likely due to syntax errors or incorrect operator usage), successful workflows demonstrate strong performance, with the best achieving 0.74 accuracy. The **middle panel** tracks the average score across generated workflows, which converges to 0.29, below the 0.46 baseline due to the high failure rate inherent in zero-shot generation. However, the **right panel** demonstrates that the best-of-$k$ selection strategy efficiently identifies high-quality solutions: the best score monotonically increases, surpassing the baseline at $k = 76$ workflows and reaching 0.74 (**60.9% improvement**) with the test set (another 100 instances) with the same trend of performance. This validates that moderate sampling suffices to discover effective workflows despite zero-shot exploration risks. The highest-scoring workflow (Listing 14 in Appendix F) implements sophisticated multi-hop reasoning through iterative retrieval-generation cycles—extracting entities, performing two-stage document search with connection analysis, and synthesizing information across retrieved contexts—demonstrating **MetaFlow**'s ability to compose complex operator sequences for unseen tools. The results confirm our meta-learning approach: train once on diverse tasks, then rapidly adapt to new operators and domains without re-optimization.

## 5.4 Main Results and Analysis

**Experimental Setup.** We evaluate **MetaFlow** on benchmarks spanning question answering (HotpotQA, DROP: 1,000 instances each), code generation (MBPP, HumanEval: full sets), and mathematical reasoning (GSM8K: 1,000 instances, MATH: Level-5 problems), following ScoreFlow's configuration with 1:4 train-test splits. We compare against two baseline categories: (1) **manually designed workflows**—IO, CoT (Wei et al., 2022), CoT SC (Wang et al., 2022), MedPrompt (Nori et al., 2023), MultiPersona (Wang et al., 2024), Self-Refine (Madaan et al., 2024); (2) **automatically optimized workflows**—ADAS (Hu et al., 2024), AFlow (Zhang et al., 2024b), ScoreFlow (Wang et al., 2025). The Planner `Qwen3-8B` uses base operators {`Generate`, `Summarize`, `Revise`, `Ensemble`} with dynamic prompt rewriting, plus novel operators {`Decompose`, `Programmer`} for OOD testing (natural language descriptions provided at inference). For each task, we generate 20 candidate workflows, validate on 50 instances, and select the best for testing. All methods use `GPT-4o-mini-0718` as executor and judge. Workflow execution is orchestrated by MetaGPT.

**Performance Analysis.** Table 1 shows **MetaFlow** achieves 78.8 average accuracy, competitive with ScoreFlow (82.5), AFlow (78.3), and ADAS (73.1) while requiring only single-inference generation versus their resource-intensive per-instance optimization. Notably, **MetaFlow** matches or

exceeds manually designed workflows and surpasses many automated methods on individual tasks (e.g., GSM8K: 93.8). The framework demonstrates strong performance across diverse domains— mathematical reasoning (GSM8K, MATH), reading comprehension (DROP), and code generation (MBPP)—validating our meta-learning approach's cross-task generalization. All baseline scores are from ScoreFlow (Wang et al., 2025). And FlowReasoner (Gao et al., 2025) achieves $82.19$ on MBPP with `GPT-4o-mini-0718` as executor.

| Method | DROP | MBPP | GSM8K | MATH | Avg |
|---|---|---|---|---|---|
| IO | 81.6 | 69.5 | 89.1 | 52.2 | 73.1 |
| CoT (Wei et al., 2022) | 83.2 | 70.4 | 88.3 | 53.4 | 73.8 |
| CoT SC (Wang et al., 2022) | 83.2 | 71.3 | 88.6 | 53.8 | 74.2 |
| MedPrompt (Nori et al., 2023) | 83.0 | 69.2 | 88.1 | 53.7 | 73.5 |
| MultiPersona (Wang et al., 2024) | 81.3 | 70.4 | 89.8 | 51.9 | 73.4 |
| Self Refine (Madaan et al., 2024) | 82.5 | 70.0 | 87.5 | 50.0 | 72.5 |
| ADAS (Hu et al., 2024) | 81.3 | 68.7 | 90.5 | 51.7 | 73.1 |
| AFlow (Zhang et al., 2024b) | 83.5 | 82.9 | 90.8 | 55.8 | 78.3 |
| ScoreFlow (Wang et al., 2025) | 86.2 | 84.7 | 94.6 | 64.4 | 82.5 |
| **Ours** | 82.8 | 77.5 | 93.8 | 61.0 | 78.8 |

Table 1: Performance comparison across multiple benchmarks. **MetaFlow** achieves 78.8 average score with single-inference generation, demonstrating competitive performance and strong cross-domain generalization.

**Understanding the Performance Gap: Paradigm Differences and Trade-offs.** Our average performance (78.8) trails ScoreFlow (82.5) by 3.7 points, which reflects fundamental differences in problem formulation rather than algorithmic limitations. This gap stems from three interacting factors. *First, risk amplification from task-level generation:* As a cross-task generator, **MetaFlow** produces one workflow $W_C$ per task—any syntax error or malformed operator call yields zero accuracy across all test instances. In contrast, ScoreFlow's instance-level approach generates $W_p$ per problem—a failed generation affects only one data point. This risk materializes in our VectorSearch experiments (Figure 4 left): 31% of generated workflows fail to execute, directly penalizing task-level metrics. *Second, the complexity-performance trade-off:* While simple workflows dominate on tasks solvable by direct reasoning, this advantage inverts for complex tool integration scenarios. On HotpotQA with the novel `VectorSearch` operator (Section 5.3), the simple CoT+RAG baseline achieves 0.46 accuracy. MetaFlow's best workflow—requiring sophisticated orchestration of retrieval, decomposition, and reasoning operators—reaches 0.74 (**+60.9%** improvement), demonstrating that complex operator composition becomes essential when tool complexity increases. Critically, even with best-of-20 validation selection, our total cost remains orders of magnitude lower than iterative search methods: 20 generations yield a reusable task-level workflow versus thousands of MCTS evaluations (AFlow) or $N$ per-instance generations (ScoreFlow). Figure 4 (right) shows best-of-$k$ scaling stabilizes around $k \approx 80$, where marginal sampling cost ($< \$1$ with `Qwen3-8B`) far outweighs other automatically designed workflows baselines.

## 6 CONCLUSION

We introduce MetaFlow, a meta-learning framework that trains language models to generate task-level workflows through a two-stage paradigm combining supervised fine-tuning with reinforcement learning across diverse task-operator combinations. Across benchmarks in question answering, code generation, and mathematical reasoning, MetaFlow achieves competitive performance (78.8 average accuracy) with single-inference generation, while demonstrating strong zero-shot generalization to novel operators and domains. The observed 31% syntax error rate in zero-shot generation highlights a key limitation of single-inference approaches—future work could explore multi-turn reinforcement learning where the model iteratively refines workflows through interaction with execution feedback, potentially combining the efficiency of learned meta-strategies with the robustness of adaptive generation.

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

# A METAFLOW ARCHITECTURE

## A.1 OPERATORS DESIGN

For the basic operators defined in the left part of Figure 2, their implementations are identical to those in the AFlow (Zhang et al., 2024b) codebase, and thus we omit their detailed descriptions here. In the following, we introduce the newly defined operators that we have designed for this work.

**Decompose Operator**: Beyond the basic text-processing operators, we introduce the Decompose operator to handle complex problem-solving scenarios that require hierarchical decomposition. This operator breaks down intricate problems into structured subproblems with explicit dependency relationships, enabling workflows to tackle multi-step reasoning tasks systematically. The operator is invoked as: await self.decompose(instruction: str, context: str) and returns a list of subproblem dictionaries, each containing an ID, description, and dependencies. The following code block shows the implementation details of this operator.

```python
class Decompose(Operator):
    """
    Core Operator: Decompose
    Breaks down complex problems into manageable subproblems.
    """
    async def __call__(self, instruction: str = "", context: str = "") -> List[Dict[str, str]]:
        prompt = f"""You are an expert at problem decomposition. Break down the complex problem into
        manageable subproblems.

**Instruction on decomposition strategy:**
{instruction}

**Problem/Context to Decompose:**
{context if context else "No context provided."}

**Original Problem:**
{self.problem_text}

Your response MUST be valid XML with 'think' and 'subproblems' fields.
- In "think", explain your decomposition strategy
- In "subproblems", provide a list where each item has:
  - id: unique identifier (e.g., "sub1", "sub2")
  - description: clear description of the subproblem
  - dependencies: comma-separated IDs of prerequisite subproblems (empty if none)

**EXAMPLE:**
<think>This problem requires finding area then volume.</think>
<subproblems>
[
  {{"id": "sub1", "description": "Calculate the radius", "dependencies": ""}},
  {{"id": "sub2", "description": "Calculate the area", "dependencies": "sub1"}},
  {{"id": "sub3", "description": "Calculate the volume", "dependencies": "sub2"}}
```

```
]
</subproblems>"""

        response = await self._fill_node(DecomposeOp, prompt, mode="xml_fill")
        response = DecomposeOp(**response)
        return [sub.dict() for sub in response.subproblems]
```

Listing 1: Implementation of the `Decompose` operator for hierarchical problem decomposition.

**SelfConsistency Operator**: To demonstrate **MetaFlow**'s capability for low-cost operator integration and continuous library expansion (Section 5.2), we introduce the `SelfConsistency` operator— a practical abstraction of the Self-Consistency reasoning pattern (Wang et al., 2022). This operator encapsulates the parallel generation with majority voting strategy, which was originally one of our experimental baselines (CoT SC in Table 1). Critically, incorporating this operator into **MetaFlow** requires *no model retraining*; the planner learns to effectively utilize it through its natural language interface description alone. The following code block presents the complete implementation:

```
class SelfConsistency(Operator):
    """Self-Consistency: multi-path sampling + majority voting for better reasoning."""

    def __init__(self, llm, problem_text: str = "", n_samples: int = 5,
                 similarity_threshold: float = 0.85, return_full_info: bool = False):
        super().__init__(llm, problem_text)
        self.n_samples = n_samples
        self.similarity_threshold = similarity_threshold
        self.return_full_info = return_full_info

    async def __call__(self, instruction: str = "", context: str = "", answer_type: str = "auto") -> str:
        silent = os.environ.get('SCOREFLOW_SILENT', 'false').lower() == 'true'
        if not silent:
            print(f"[SelfConsistency] samples={self.n_samples}, threshold={self.similarity_threshold}")

        paths = await self._parallel_sampling(instruction, context)
        if not silent:
            print(f"  Generated {len(paths)} paths")

        raw_answers = await self._extract_answers_batch(paths, answer_type)
        if not silent:
            print(f"  Extracted {len([a for a in raw_answers if a])} answers")

        normalized = [self._normalize_answer(a, answer_type) for a in raw_answers]
        clustered = self._cluster_similar_answers(normalized)
        votes = Counter(clustered)

        if not votes:
            if not silent:
                print("  No valid answers, fallback")
            return await self._fallback_generate(instruction, context)

        answer, count = votes.most_common(1)[0]
        conf = count / len(clustered) if clustered else 0
        if not silent:
            print(f"  Votes: {dict(votes)} | Answer: {answer} | Conf: {conf:.1%}")

        if self.return_full_info:
            return self._format_full_response(paths, raw_answers, votes, answer, conf)
        return answer

    async def _parallel_sampling(self, instruction: str, context: str) -> List[str]:
        prompt = f"""Solve step by step.
**Problem:** {self.problem_text}
**Instruction:** {instruction or "Solve carefully."}
**Context:** {context or "None"}
Show reasoning. End with "Final Answer:" or "The answer is:"
**Solution:**"""
        tasks = [self._sample_single_path(prompt) for _ in range(self.n_samples)]
        results = await asyncio.gather(*tasks, return_exceptions=True)
        return [r for r in results if isinstance(r, str) and len(r.strip()) > 10]

    async def _sample_single_path(self, prompt: str) -> str:
        try:
            return (await self._fill_node(GenerateOp, prompt, mode="single_fill")).get("response", "")
        except:
            return ""

    async def _extract_answers_batch(self, paths: List[str], answer_type: str) -> List[str]:
        answers = []
        for p in paths:
            ans = self._rule_based_extraction(p, answer_type)
            answers.append(ans if ans else await self._llm_extract_answer(p, answer_type))
        return answers

    def _rule_based_extraction(self, text: str, answer_type: str) -> str:
        patterns = [
            r"(?:final answer|the answer is|answer:)\s*[:\s]*(.+?)(?:\n|$|\.(?:\s|$))",
            r"(?:therefore|thus|so|hence),?\s*(?:the answer is)?\s*[:\s]*(.+?)(?:\n|$|\.(?:\s|$))",
            r"\\boxed\{(.+?)\}", r"\*\*(.+?)\*\*\s*$",
        ]
```

```
        for p in patterns:
            m = re.search(p, text.lower(), re.IGNORECASE | re.MULTILINE)
            if m:
                ans = re.sub(r'^[:\s]+|[:\s\.]+$', '', m.group(1).strip())
                if ans and len(ans) < 200:
                    return ans
        if answer_type == "choice":
            m = re.search(r"(?:answer|option|choice)[:\s]*([A-Da-d])\b", text, re.IGNORECASE)
            if m: return m.group(1).upper()
        if answer_type == "numeric":
            nums = re.findall(r'-?\d+(?:\.\d+)?(?:/\d+)?', text)
            if nums: return nums[-1]
        return ""

    async def _llm_extract_answer(self, path: str, answer_type: str) -> str:
        hints = {"numeric": "number only", "choice": "letter only", "short": "brief", "boolean": "yes/no
        only"}
        prompt = f"Extract final answer. {hints.get(answer_type, '')} Return ONLY the answer.\n**Solution
        :** {path[-2000:]}\n**Answer:**"
        try:
            ans = (await self._fill_node(GenerateOp, prompt, mode="single_fill")).get("response", "").strip
            ()
            return re.sub(r'^(the answer is|answer:|final answer:)\s*', '', ans, flags=re.IGNORECASE).strip
            ()
        except:
            return ""

    def _normalize_answer(self, answer: str, answer_type: str) -> str:
        if not answer: return ""
        n = re.sub(r'^(the |a |an )', '', answer.strip().lower())
        n = re.sub(r'[.,!?;:]+$', '', n)
        if answer_type == "numeric" or re.match(r'^-?\d', n):
            n = re.sub(r'[$%,\s]', '', n)
            if '/' in n:
                try: p = n.split('/'); n = str(float(p[0]) / float(p[1])) if len(p) == 2 else n
                except: pass
        elif answer_type == "choice":
            m = re.search(r'[A-Da-d]', n)
            if m: n = m.group(0).upper()
        elif answer_type == "boolean":
            n = 'yes' if n in ['yes','true','1','correct','right'] else ('no' if n in ['no','false','0','
            incorrect','wrong'] else n)
        return n

    def _cluster_similar_answers(self, answers: List[str]) -> List[str]:
        valid = [a for a in answers if a]
        if not valid or self.similarity_threshold >= 1.0: return valid
        clusters = []
        for ans in valid:
            matched = False
            for rep, members in clusters:
                if SequenceMatcher(None, ans, rep).ratio() >= self.similarity_threshold:
                    members.append(ans); matched = True; break
            if not matched: clusters.append((ans, [ans]))
        return [Counter(m).most_common(1)[0][0] for a in valid for r, m in clusters if a in m]

    async def _fallback_generate(self, instruction: str, context: str) -> str:
        prompt = f"Solve: {self.problem_text}\nInstruction: {instruction}\nContext: {context or 'None'}"
        return (await self._fill_node(GenerateOp, prompt, mode="single_fill")).get("response", "No answer")

    def _format_full_response(self, paths: List[str], raw: List[str], votes: Counter, answer: str, conf:
    float) -> str:
        lines = [f"## Self-Consistency Result", f"**Final Answer:** {answer}", f"**Confidence:** {conf:.1%}
        ",
                 "### Votes"] + [f"- {a}: {c}" for a, c in votes.most_common()]
        return "\n".join(lines)
```

Listing 2: Implementation of the SelfConsistency operator, which abstracts the Self-Consistency pattern (Wang et al., 2022) into a reusable component. This operator performs parallel sampling of multiple reasoning paths (default $n = 5$) and applies majority voting with similarity-based answer clustering to select the most consistent solution. The implementation demonstrates **MetaFlow**'s low-cost extensibility: practitioners can encapsulate manually-tested reasoning patterns into operators without model retraining.

**VectorSearch Operator**: To evaluate the generalization capability of **MetaFlow** to completely out-of-distribution (OOD) operators, we introduce the VectorSearch operator 3, which is a complex tool-calling operator that was entirely unseen during both the SFT and RLVR training phases. Unlike the basic text-processing operators used in training, VectorSearch requires sophisticated external API interactions with a vector database (ChromaDB) and involves multiple parameters for controlling retrieval behavior. The operator is invoked as: await self.vector_search(instruction: str, context: str, top_k: int), where it performs hybrid retrieval combining document-level and sentence-level semantic search over the HotpotQA knowledge base. Despite its complexity and complete absence from training data, **MetaFlow** successfully learns to incorporate this operator into

generated workflows based solely on its natural language description provided at inference time, as demonstrated in Appendix F.

```python
class VectorSearch(Operator):
    """
    Core Operator: Vector Search
    Retrieves relevant documents from HotpotQA vector database using RAG system.
    """

    def __init__(self, llm, problem_text: str = "", db_config: Dict = None):
        super().__init__(llm, problem_text)
        self.config = self._load_config()
        if db_config:
            self.config.update(db_config)
        self._init_chromadb()

    def _load_config(self) -> Dict:
        """Load configuration from db.config file"""
        from pathlib import Path
        config = {}
        config_file = Path(__file__).parent / "db.config"

        if config_file.exists():
            with open(config_file, 'r') as f:
                for line in f:
                    line = line.strip()
                    if line and not line.startswith('#') and '=' in line:
                        key, value = line.split('=', 1)
                        key, value = key.strip(), value.strip()
                        if key in ('DOC_TOP_K', 'SENT_TOP_K'):
                            config[key.lower()] = int(value)
                        elif key == 'HYBRID_MODE':
                            config[key.lower()] = value.lower() == 'true'
                        else:
                            config[key.lower()] = value
        else:
            config = {
                'db_path': '...chroma_db', 'model_path': '...all-MiniLM-L6-v2',
                'doc_top_k': 3, 'sent_top_k': 5, 'hybrid_mode': True
            }
        return config

    def _init_chromadb(self):
        """Initialize ChromaDB connection and collections"""
        try:
            import chromadb
            from chromadb.utils import embedding_functions
            self.client = chromadb.PersistentClient(path=self.config['db_path'])
            model = self.config['model_path'] if os.path.exists(self.config['model_path']) else "all-MiniLM-L6-v2"
            self.embedding_function = embedding_functions.SentenceTransformerEmbeddingFunction(model_name=model)
            self.doc_collection = self.client.get_collection("documents")
            self.sent_collection = self.client.get_collection("sentences")
        except Exception as e:
            self.doc_collection = self.sent_collection = None

    async def __call__(self, instruction: str = "", context: str = "", top_k: int = None) -> str:
        """Execute vector search and return formatted retrieved documents."""
        if not self.doc_collection or not self.sent_collection:
            return "Error: Vector database not initialized."
        doc_k = top_k or self.config['doc_top_k']
        query = await self._process_query(instruction, context)
        retrieved_data = self._hybrid_retrieval(query, doc_k, self.config['sent_top_k'])
        return self._format_context(retrieved_data)

    async def _process_query(self, instruction: str, context: str) -> str:
        """Process and combine query from instruction and context."""
        if instruction and context:
            return f"{instruction} {context}"
        return instruction or (context[:200] if context else self.problem_text[:200] or "general information")

    def _hybrid_retrieval(self, query: str, doc_k: int, sent_k: int) -> Dict:
        """Perform hybrid document and sentence level retrieval."""
        retrieved_data = {'documents': [], 'scores': []}
        try:
            if self.config.get('hybrid_mode', True):
                doc_results = self.doc_collection.query(query_texts=[query], n_results=doc_k)
                for i, (doc_id, doc_text, meta, dist) in enumerate(zip(
                        doc_results['ids'][0], doc_results['documents'][0],
                        doc_results['metadatas'][0], doc_results['distances'][0])):
                    retrieved_data['documents'].append({
                        'doc_id': doc_id, 'title': meta.get('title', 'Unknown'),
                        'text': doc_text[:500], 'type': 'document', 'rank': i + 1})
                    retrieved_data['scores'].append(float(dist))
            sent_results = self.sent_collection.query(query_texts=[query], n_results=sent_k)
            for i, (sid, stxt, meta, dist) in enumerate(zip(
                    sent_results['ids'][0], sent_results['documents'][0],
                    sent_results['metadatas'][0], sent_results['distances'][0])):
                if i < 3:
                    retrieved_data['documents'].append({
```

```
                         'doc_id': sid, 'title': meta.get('title', 'Unknown'),
                         'text': stxt, 'type': 'sentence', 'rank': i + 1})
                    retrieved_data['scores'].append(float(dist))
        except Exception:
            pass
        return retrieved_data

    def _format_context(self, retrieved_data: Dict) -> str:
        """Format retrieved documents into readable context string."""
        if not retrieved_data['documents']:
            return "No relevant documents found."
        parts = ["**Retrieved Information:**\n"]
        for d in retrieved_data['documents']:
            parts.append(f"[{d['type'].title()}: {d['title']}] {d['text']}\n")
        return "\n".join(parts)
```

Listing 3: Implementation of the VectorSearch operator for retrieval-augmented generation.

## A.2 INPUT AND OUTPUT OF THE MODEL

The MODEL INPUT consists of two parts: (1) TASK TYPE DESCRIPTION, which elucidates the domain characteristics of the target problem family and the input-output formats of each belonging problem instance. The block below shows the complete TASK TYPE DESCRIPTION for GSM8K.

```
### Problem Domain Overview
This domain tests multi-step mathematical reasoning using basic arithmetic operations.

#### Key Characteristics & Requirements
- **Answer Format**: Single numerical value (integer or decimal)
- **Solution Steps**: 2-8 step reasoning chains using +, -, *, /
- **Critical**: Track intermediate results and units throughout
- **Validation**: Final answer must be numerically exact
- **No Complex Math**: Only elementary arithmetic, no algebra or calculus

#### Common Problem Types & Solution Strategies
- **Sequential Operations**: Step-by-step calculations building on previous results
- **Rate Problems**: Distance/speed/time, work rates, unit prices
- **Distribution**: Dividing quantities, equal sharing, remainders
- **Proportions**: Percentages, fractions, ratios, scaling
- **Multi-entity**: Track different quantities for multiple people/objects

#### Workflow Focus Points
1. Extract all numerical values and their context
2. Identify what the question asks for
3. Build step-by-step calculation chain
4. Show intermediate results explicitly
5. Return final numerical answer only

#### Input Format
```
---
**QUESTION:**
[Complete word problem text]
---
```
Multiple problems follow the same structure if provided.
```

Listing 4: Complete TASK TYPE DESCRIPTION of GSM8K including domain overview, key requirements and the format of problem instance.

(2) OPERATOR DESCRIPTIONS, which define the functions, parameters, and input-output formats of each available operator (whether pre-set or user-defined). The block below shows the complete OPERATOR DESCRIPTIONS of the operator set Ops = (Generate, Summarize, Revise, Ensemble).

```
### Available Operators & Building Blocks
All operators follow a consistent interface pattern and are initialized with the problem text.

#### **Core Operators**
**Generate: CREATE new information**
- **Signature:** `await self.generate(instruction: str, context: str = "") -> str`
- **Purpose:** Produces new text, analysis, or reasoning based on strategic instructions

**Revise: IMPROVE existing information**
- **Signature:** `await self.revise(instruction: str, context: str) -> str`
- **Purpose:** Critiques and refines existing text based on specific improvement criteria

**Summarize: COMPRESS information**
- **Signature:** `await self.summarize(instruction: str, context: str) -> str`
- **Purpose:** Condenses text while preserving key information relevant to the problem

**Ensemble: DECIDE between or synthesize options**
- **Signature:** `await self.ensemble(instruction: str, contexts: List[str]) -> str`
- **Purpose:** Evaluates, compares, or merges multiple candidate solutions
```

Listing 5: Complete OPERATOR DESCRIPTIONS of an operator set.

The OUTPUT requires a structured WORKFLOW aimed at efficiently solving all problem instances under the task type using the given operator set based on the template below.

```python
# --- DO NOT IMPORT HERE ---
class Workflow:
    def __init__(self, config, problem) -> None:
        # --- DO NOT MODIFY THIS SECTION ---
        self.config = config
        self.problem_text = problem
        self.llm = create(config)

        self.generate = operator.Generate(self.llm, self.problem_text)
        self.revise = operator.Revise(self.llm, self.problem_text)
        self.summarize = operator.Summarize(self.llm, self.problem_text)
        self.ensemble = operator.Ensemble(self.llm, self.problem_text)

    async def run_workflow(self):
        """
        Implement the core problem-solving logic here.
        """
        import asyncio
        # --- YOUR WORKFLOW LOGIC HERE ---
```

Listing 6: OUTPUT template of given Ops = (Generate, Summarize, Revise, Ensemble).

## B  DETAILS OF SUPERVISED FINE-TUNING

The supervised fine-tuning (SFT) stage initializes the planner model $\pi_\theta$ to generate syntactically correct workflows following the required template structure (see Appendix A.2 for the detailed input-output format). This stage addresses the cold start problem by teaching the model the basic grammar of workflow construction before reinforcement learning optimization.

**Dataset Construction Pipeline.** We construct approximately 1,300 high-quality ([C, Ops], W) pairs using Qwen-Max as the expert model. The construction follows a four-stage pipeline:

1. **Prompt Crafting**: Construct input prompts combining task descriptions C (domain characteristics, input-output formats) and operator specifications Ops (function signatures, purposes) following the format defined in Appendix A.2.

2. **Expert Generation**: Feed prompts to Qwen-Max API to obtain comprehensive responses including workflow explanations and complete executable code.

3. **Code Extraction**: Parse API responses to extract clean workflow code W, discarding natural language explanations.

4. **Quality Verification**: Execute each extracted workflow on 1-2 randomly sampled problem instances $p \sim C$ to ensure (a) no syntax errors and (b) correct solutions. Only validated workflows are retained.

This pipeline is applied across four tasks (GSM8K, DROP, MBPP, Humaneval) with the basic operator set $\text{Ops}^{\text{SFT}} = (\text{Generate}, \text{Summarize}, \text{Revise}, \text{Ensemble})$, producing diverse workflow patterns that establish the foundation for RLVR optimization. To ensure stable training on this limited dataset, we employ parameter-efficient fine-tuning with LoRA configuration, as detailed in Section B.2.

### B.1  DATASET CONSTRUCTION EXAMPLES

To illustrate the construction pipeline, we present a concrete example for the GSM8K mathematical reasoning task. The input prompt (Listing 7) combines task description and operator specifications:

```
### 1. Problem Domain Overview
This domain tests multi-step mathematical reasoning using basic arithmetic operations.

#### Key Characteristics & Requirements
- **Answer Format**: Single numerical value (integer or decimal)
- **Solution Steps**: 2-8 step reasoning chains using +, -, *, /
- **Critical**: Track intermediate results and units throughout
- **Validation**: Final answer must be numerically exact
- **No Complex Math**: Only elementary arithmetic, no algebra or calculus

#### Common Problem Types & Solution Strategies
- **Sequential Operations**: Step-by-step calculations building on previous results
```

```
972
973    - **Rate Problems**: Distance/speed/time, work rates, unit prices
974    - **Distribution**: Dividing quantities, equal sharing, remainders
       - **Proportions**: Percentages, fractions, ratios, scaling
       - **Multi-entity**: Track different quantities for multiple people/objects
975
976    #### Workflow Focus Points
       1. Extract all numerical values and their context
977    2. Identify what the question asks for
       3. Build step-by-step calculation chain
978    4. Show intermediate results explicitly
       5. Return final numerical answer only
979
980    ### 2. Available Operators & Building Blocks
       All operators follow a consistent interface pattern and are initialized with the problem text.
981
982    #### **Core Operators**
       **Generate: CREATE new information**
983    - **Signature:** `await self.generate(instruction: str, context: str = "") -> str`
       - **Purpose:** Produces new text, analysis, or reasoning based on strategic instructions
984
985    **Revise: IMPROVE existing information**
       - **Signature:** `await self.revise(instruction: str, context: str) -> str`
986    - **Purpose:** Critiques and refines existing text based on specific improvement criteria
987    **Summarize: COMPRESS information**
       - **Signature:** `await self.summarize(instruction: str, context: str) -> str`
988    - **Purpose:** Condenses text while preserving key information relevant to the problem
989    **Ensemble: DECIDE between or synthesize options**
       - **Signature:** `await self.ensemble(instruction: str, contexts: List[str]) -> str`
990    - **Purpose:** Evaluates, compares, or merges multiple candidate solutions
991
992    #### Input Format
       ```
993    ---
       **QUESTION:**
994    [Complete word problem text]
       ---
995    ```
       Multiple problems follow the same structure if provided.
996
997    ### 3. Your Task: Complete the `run_workflow` Method
       Your task is to write the Python code for the `run_workflow` method within the provided template.
998    Focus on creating a robust, reusable workflow that leverages detailed instructions.
999    **Base Template:**
       ```python
1000   # --- DO NOT IMPORT HERE ---
       class Workflow:
1001       def __init__(self, config, problem) -> None:
1002           # --- DO NOT MODIFY THIS SECTION ---
               self.config = config
1003           self.problem_text = problem
               self.llm = create(config)
1004
1005           self.generate = operator.Generate(self.llm, self.problem_text)
               self.revise = operator.Revise(self.llm, self.problem_text)
1006           self.summarize = operator.Summarize(self.llm, self.problem_text)
               self.ensemble = operator.Ensemble(self.llm, self.problem_text)
1007
1008       async def run_workflow(self):
               """
1009           Implement the core problem-solving logic here.
               """
1010           import asyncio
               # --- YOUR WORKFLOW LOGIC HERE ---
1011   ```
1012
1013
```

Listing 7: Input prompt for `Qwen-Max` to generate workflow examples (GSM8K task).

Given this input prompt, `Qwen-Max` generates a comprehensive response that includes both natural language explanation of the workflow design and the complete workflow code. Listing 8 shows the raw output from the API:

```
1017   Now let me understand the task better and implement the `run_workflow` method.
1018   This is a workflow for solving multi-step mathematical reasoning problems using the provided operators.

1019   I've created a robust workflow for multi-step mathematical reasoning problems. Here's how it works:

1020   ```python
       # --- DO NOT IMPORT HERE ---
1021   class Workflow:
           def __init__(self, config, problem) -> None:
1022           # --- DO NOT MODIFY THIS SECTION ---
               self.config = config
1023           self.problem_text = problem
               self.llm = create(config)
1024
1025           self.generate = operator.Generate(self.llm, self.problem_text)
               self.revise = operator.Revise(self.llm, self.problem_text)
```

```
        self.summarize = operator.Summarize(self.llm, self.problem_text)
        self.ensemble = operator.Ensemble(self.llm, self.problem_text)

    async def run_workflow(self):
        """
        Implement the core problem-solving logic here.
        """
        # Step 1: Extract all numerical values and their context
        extracted_info = await self.generate(
            instruction="Extract all numerical values from the problem, including their units and
            contextual meaning (e.g., '5 apples', 'speed of 60 km/h'). List them clearly.",
            context=self.problem_text
        )

        # Step 2: Identify what the question asks for
        target_query = await self.generate(
            instruction="Based on the problem, state precisely what numerical value is being asked for,
            including its expected unit if any.",
            context=self.problem_text
        )

        # Step 3: Build a step-by-step calculation plan
        plan = await self.generate(
            instruction="Create a clear, step-by-step plan (2 to 8 steps) to solve the problem using only
            +, -, *, / operations. Reference the extracted numbers and specify what each step computes.",
            context=f"Extracted info: {extracted_info}\nTarget: {target_query}"
        )

        # Step 4: Execute the plan with explicit intermediate results
        calculation = await self.generate(
            instruction="Perform the arithmetic step-by-step as per the plan. Show each intermediate result
             with its unit or meaning. Ensure only elementary arithmetic is used.",
            context=plan
        )

        # Step 5: Extract and validate the final numerical answer
        final_answer_raw = await self.generate(
            instruction="From the calculation steps, extract only the final numerical answer (as an integer
             or decimal). Do not include units, explanations, or text -- only the number.",
            context=calculation
        )

        # Step 6: Revise to ensure numerical exactness and proper format
        final_answer_clean = await self.revise(
            instruction="Ensure the output is a single numerical value (e.g., 42 or 15.75), with no extra
            characters, units, or formatting. If it's a whole number, do not include decimal places unless
            required.",
            context=final_answer_raw
        )

        # Step 7: Return the final answer
        return final_answer_clean.strip()
```

**Workflow Design:**

1. **Problem Extraction** - Uses `generate` to parse the problem, extracting all numerical values, the goal
, and relationships between quantities

2. **Parallel Solution Generation** - Generates three independent solutions using different approaches:
   - Standard step-by-step forward calculation
   - Alternative verification method
   - Working backwards from the goal

3. **Solution Revision** - Each solution is revised to catch arithmetic errors, verifying each calculation
step

4. **Ensemble Decision** - Compares all three verified solutions to determine the correct answer through
majority vote or error analysis

5. **Answer Extraction** - Uses `summarize` to extract just the final numerical value

**Key Features:**
- Uses `asyncio.gather` for parallel execution (generating and revising solutions simultaneously)
- Multiple independent solutions reduce the chance of systematic errors
- Revision step catches arithmetic mistakes
- Ensemble aggregates results for higher accuracy
- Clean numerical output extraction at the end

Listing 8: Raw output from `Qwen-Max` API containing workflow explanation and code.

Finally, we parse the API response to extract only the executable workflow code, discarding the natural language explanations. This extracted code forms a single training example in our SFT dataset. Listing 9 shows the clean workflow code extracted from the above output:

```
# --- DO NOT IMPORT HERE ---
class Workflow:
    def __init__(self, config, problem) -> None:
        # --- DO NOT MODIFY THIS SECTION ---
        self.config = config
```

```
        self.problem_text = problem
        self.llm = create(config)

        self.generate = operator.Generate(self.llm, self.problem_text)
        self.revise = operator.Revise(self.llm, self.problem_text)
        self.summarize = operator.Summarize(self.llm, self.problem_text)
        self.ensemble = operator.Ensemble(self.llm, self.problem_text)

    async def run_workflow(self):
        """
        Implement the core problem-solving logic here.
        """
        # Step 1: Extract all numerical values and their context
        extracted_info = await self.generate(
            instruction="Extract all numerical values from the problem, including their units and
            contextual meaning (e.g., '5 apples', 'speed of 60 km/h'). List them clearly.",
            context=self.problem_text
        )

        # Step 2: Identify what the question asks for
        target_query = await self.generate(
            instruction="Based on the problem, state precisely what numerical value is being asked for,
            including its expected unit if any.",
            context=self.problem_text
        )

        # Step 3: Build a step-by-step calculation plan
        plan = await self.generate(
            instruction="Create a clear, step-by-step plan (2 to 8 steps) to solve the problem using only
            +, -, *, / operations. Reference the extracted numbers and specify what each step computes.",
            context=f"Extracted info: {extracted_info}\nTarget: {target_query}"
        )

        # Step 4: Execute the plan with explicit intermediate results
        calculation = await self.generate(
            instruction="Perform the arithmetic step-by-step as per the plan. Show each intermediate result
             with its unit or meaning. Ensure only elementary arithmetic is used.",
            context=plan
        )

        # Step 5: Extract and validate the final numerical answer
        final_answer_raw = await self.generate(
            instruction="From the calculation steps, extract only the final numerical answer (as an integer
             or decimal). Do not include units, explanations, or text -- only the number.",
            context=calculation
        )

        # Step 6: Revise to ensure numerical exactness and proper format
        final_answer_clean = await self.revise(
            instruction="Ensure the output is a single numerical value (e.g., 42 or 15.75), with no extra
            characters, units, or formatting. If it's a whole number, do not include decimal places unless
            required.",
            context=final_answer_raw
        )

        # Step 7: Return the final answer
        return final_answer_clean.strip()
```

Listing 9: Extracted workflow code for SFT training data.

The above example demonstrates the complete four-stage pipeline from input prompt to validated workflow code, illustrating how each training pair is constructed and verified.

## B.2 LoRA Configuration

To mitigate the risk of mode collapse when fine-tuning on this limited dataset of approximately 1,300 examples, we employ Low-Rank Adaptation (LoRA) with rank-16 (Hu et al., 2022) and train for one epoch with batch size of 16, as illustrated in Figure 3. This parameter-efficient approach serves as an effective regularization mechanism: preliminary experiments with full-parameter fine-tuning resulted in degenerate repetition patterns (Listing 10), where the model generates circular, non-terminating reasoning loops instead of producing valid workflows.

```
To solve this problem, we need to first understand what the problem is asking. The problem is asking us to
find the answer. To find the answer, we need to solve this problem. To solve this problem, we need to first
 understand what the problem is asking. The problem is asking us to find the answer. To find the answer, we
 need to solve this problem. To solve this problem, we need to first understand what the problem is asking.
 The problem is asking us to find the answer. To find the answer, we need to solve this problem...
```

Listing 10: Example of degenerate repetition pattern (mode collapse) observed during full-parameter fine-tuning on the limited SFT dataset.

## C   DETAILS OF REINFORCEMENT LEARNING WITH VERIFIABLE REWARDS

The task set remains the same as the SFT stage:

$$\mathcal{D}_{\text{train}}^{\text{RLVR}} = \mathcal{D}_{\text{train}}^{\text{SFT}} = \{\text{GSM8K, DROP, MBPP, Humaneval}\}$$

To enhance the model's generalization to diverse operator configurations, we train with four different operator sets that progressively introduce novel operators beyond the base SFT set:

$$\mathsf{Ops}^{\text{RLVR},1} = (\mathsf{Generate}, \mathsf{Summarize}, \mathsf{Revise}, \mathsf{Ensemble})$$

$$\mathsf{Ops}^{\text{RLVR},2} = (\mathsf{Generate}, \mathsf{Summarize}, \mathsf{Revise}, \mathsf{Ensemble}, \mathsf{Programmer})$$

$$\mathsf{Ops}^{\text{RLVR},3} = (\mathsf{Generate}, \mathsf{Summarize}, \mathsf{Revise}, \mathsf{Ensemble}, \mathsf{Decompose})$$

$$\mathsf{Ops}^{\text{RLVR},4} = (\mathsf{Generate}, \mathsf{Summarize}, \mathsf{Revise}, \mathsf{Ensemble}, \mathsf{Programmer}, \mathsf{Decompose})$$

During each training iteration, we randomly sample one operator set from $\{\mathsf{Ops}^{\text{RLVR},i}\}_{i=1}^4$ along with a task from $\mathcal{D}_{\text{train}}^{\text{RLVR}}$, forming diverse domain combinations $(\mathsf{C}, \mathsf{Ops})$ for meta-learning. This diversity ensures the planner learns operator-agnostic workflow construction principles rather than memorizing fixed operator-task associations.

## D   CASE STUDY: WORKFLOW DESIGN TRANSFORMATION AFTER TRAINING

### D.1   OVERVIEW

To better understand the effectiveness of our training paradigm, we conduct a critical ablation study analyzing changes in model behavior before and after training. We compare two randomly selected workflows generated by the untrained base model (Qwen3-8B) versus the fully trained **MetaFlow** model after our two-stage training.

### D.2   IMPACT OF TRAINING ON WORKFLOW GENERATION BEHAVIOR

**Before Training (Base Model Qwen3-8B):** Without SFT and RLVR optimization, the base model generates workflows with multi-step logic following a linear process: extract function name $\rightarrow$ summarize $\rightarrow$ generate $\rightarrow$ revise $\rightarrow$ ensemble.

```python
class Workflow:
    # ... (initialization code omitted) ...
    async def run_workflow(self):
        # Step 1: Extract function name from test cases
        func_name = await self.generate(
            instruction="Extract the function name from the test cases.
                        Return ONLY the function name, nothing else.",
            context=self.problem_text
        )
        # Step 2: Summarize problem description and test cases
        problem_summary = await self.summarize(
            instruction="Summarize the problem description and test cases
                        to create a clear problem statement.",
            context=self.problem_text
        )
        # Step 3: Generate code based on problem summary
        generated_code = await self.generate(
            instruction=f"Write a Python function named '{func_name}'...
                        Return ONLY the function definition, nothing else.",
            context=problem_summary
        )
        # Steps 4 & 5: Revise and ensemble
        revised_code = await self.revise(...)
        final_code = await self.ensemble(contexts=[generated_code, revised_code])
        return final_code
```

Listing 11: Workflow generated by the untrained base model.

**Analysis:** While structured, this workflow has potential weaknesses:

1. **Information Bottleneck:** The summarize step, intended to condense information, risks losing critical details or constraints from the original problem description.

2. **Instruction Ambiguity:** Vague instructions like "Return ONLY the function definition" can be misinterpreted, leading to incomplete code that lacks necessary import statements or helper functions.

**After Training (MetaFlow Model):** In contrast, the **MetaFlow** model optimized through our two-stage training paradigm generates a workflow that, while maintaining complexity, is significantly more robust and methodical in its approach, and has learned the self-consistency operator unseen during training.

```python
% --- DO NOT IMPORT HERE ---
class Workflow:
    def __init__(self, config, problem) -> None:
        # --- DO NOT MODIFY THIS SECTION ---
        self.config = config
        self.problem_text = problem
        self.llm = create(config)

        self.generate = operator.Generate(self.llm, self.problem_text)
        self.revise = operator.Revise(self.llm, self.problem_text)
        self.summarize = operator.Summarize(self.llm, self.problem_text)
        self.ensemble = operator.Ensemble(self.llm, self.problem_text)
        self.selfconsistency = operator.SelfConsistency(self.llm, self.problem_text)

    async def run_workflow(self):
        """
        Implement the core problem-solving logic here.
        Remember:
        - Use detailed, comprehensive instructions
        - Dynamic instruction construction is powerful
        - All operators expect (instruction: str, context: str) except Ensemble which takes contexts: List[
        str]
        """
        import asyncio

        # Step 1: Extract function name from test cases
        function_name = await self.generate(
            instruction="""
            Analyze the test cases provided and extract the function name.
            The function name should match the naming convention used in the test cases.
            Return ONLY the function name, nothing else.
            """,
            context=self.problem_text
        )

        # Step 2: Extract input types and expected outputs from test cases
        input_types_output = await self.generate(
            instruction="""
            Analyze the test cases provided and extract the input types and expected outputs.
            Format the response as a dictionary where keys are the parameter names and values are the types
            .
            Also, identify the expected output format.
            Return ONLY the structured information, nothing else.
            """,
            context=self.problem_text
        )

        # Step 3: Generate multiple candidate implementations
        implementations = await asyncio.gather(
            self.generate(
                instruction=f"""
                Given the problem description: {self.problem_text}
                Generate a Python function that satisfies the test cases.
                Ensure that the function name is '{function_name}'.
                Include all necessary imports at the beginning of the function.
                Use the extracted input types and expected outputs: {input_types_output}.
                """,
                context=self.problem_text
            ),
            # ... (additional parallel generate calls omitted for brevity) ...
            self.generate(
                instruction=f"""
                Given the problem description: {self.problem_text}
                Generate another Python function that satisfies the test cases.
                Ensure that the function name is '{function_name}'.
                Include all necessary imports at the beginning of the function.
                Use the extracted input types and expected outputs: {input_types_output}.
                """,
                context=self.problem_text
            )
        )

        # Step 4: Select the best implementation using SelfConsistency
        best_implementation = await self.selfconsistency(
            instruction=f"""
            Evaluate the following candidate implementations:
            {implementations[0]}
            {implementations[1]}

            Select the best implementation based on the following criteria:
```

```
                - Consistency with the test cases
                - Logical correctness
                - Efficient and clean code
                - Proper use of imports and function signature

                Return ONLY the best implementation, nothing else.
                """,
                context=self.problem_text
        )

        return best_implementation
```

Listing 12: Workflow generated by the trained **MetaFlow** model.

**Analysis:** The trained model's workflow exhibits several key improvements contributing to its robustness:

1. **From Compression to Structured Extraction:** The model avoids the potentially lossy `summarize` step. Instead, it performs targeted extractions to parse unstructured text into structured data (e.g., function name, I/O types). This approach preserves information integrity and provides a more reliable foundation for subsequent steps.

2. **Parallel Exploration for Increased Robustness:** Rather than following a single generation-and-revision path, the model generates multiple, independent candidate solutions in parallel using `asyncio.gather`. This multi-path exploration inherently increases the robustness of the process by diversifying the solution space and reducing the risk of settling on a single, flawed implementation.

3. **More Explicit and Context-Aware Instructions:** The instructions are dynamically constructed using f-strings to include context from prior steps (e.g., 'function_name', 'input_types_output'). This makes the prompts highly specific and unambiguous, directly mitigating the weaknesses of the base model and ensuring that each generative step is precisely guided.

# E   SELF CONSISTENCY OPERATOR: LOW-COST INTEGRATION EXAMPLE

This section demonstrates **MetaFlow**'s ability to immediately generate effective workflows with newly integrated operators without retraining. After introducing the `SelfConsistency` operator (which encapsulates parallel generation with majority voting), the model generates the following workflow for GSM8K mathematical reasoning through a single inference.

```
# --- DO NOT IMPORT HERE ---
class Workflow:
    def __init__(self, config, problem) -> None:
        # --- DO NOT MODIFY THIS SECTION ---
        self.config = config
        self.problem_text = problem
        self.llm = create(config)

        self.generate = operator.Generate(self.llm, self.problem_text)
        self.revise = operator.Revise(self.llm, self.problem_text)
        self.summarize = operator.Summarize(self.llm, self.problem_text)
        self.ensemble = operator.Ensemble(self.llm, self.problem_text)
        self.self_consistency = operator.SelfConsistency(
            self.llm,
            self.problem_text,
            n_samples=5,
            similarity_threshold=0.9,
            return_full_info=False
        )

    async def run_workflow(self):
        """
        Implement the core problem-solving logic here.
        Remember:
        - Use detailed, comprehensive instructions
        - Dynamic instruction construction is powerful
        - All operators expect (instruction: str, context: str) except Ensemble which takes contexts: List[
        str]
        """
        import asyncio
        sc_instruction = (
            "Solve this math problem step by step. "
            "Show all calculations clearly. "
            "At the end, state your final numerical answer."
        )
```

```
        sc_answer = await self.self_consistency(
            instruction=sc_instruction,
            context="",
            answer_type="numeric"
        )

        format_instruction = (
            f"The calculated answer is: {sc_answer}\n\n"
            "Extract ONLY the final numerical answer. "
            "If it's a decimal, round to a reasonable precision. "
            "If the answer represents a count of items/people, it should be a whole number. "
            "Return ONLY the number, nothing else."
        )

        final_answer = await self.generate(
            instruction=format_instruction,
            context=sc_answer
        )

        return final_answer.strip()
```

Listing 13: Example workflow generated by **MetaFlow** for GSM8K after integrating the SelfConsistency operator. The workflow demonstrates immediate operator utilization: (1) invoking SelfConsistency with n=5 parallel samples and similarity-based voting, and (2) formatting the numeric answer. This workflow was generated without any model retraining, showcasing low-cost operator extensibility.

## F CASE STUDY: ZERO-SHOT WORKFLOW GENERATION WITH NOVEL VECTORSEARCH OPERATOR

This case study demonstrates **MetaFlow**'s zero-shot generalization to the novel VectorSearch operator on HotpotQA multi-hop question answering. The operator was never encountered during SFT or RLVR training, yet the model successfully generates workflows that orchestrate complex retrieval-reasoning pipelines. Listing 14 shows the highest-scoring workflow (0.74 accuracy) among 100 generated candidates, which achieves 60.9% improvement over the CoT+RAG baseline (0.46 accuracy).

```
# --- DO NOT IMPORT HERE ---
class Workflow:
    def __init__(self, config, problem) -> None:
        # --- DO NOT MODIFY THIS SECTION ---
        self.config = config
        self.problem_text = problem
        self.llm = create(config)

        self.generate = operator.Generate(self.llm, self.problem_text)
        self.revise = operator.Revise(self.llm, self.problem_text)
        self.summarize = operator.Summarize(self.llm, self.problem_text)
        self.ensemble = operator.Ensemble(self.llm, self.problem_text)
        self.vector_search = operator.VectorSearch(self.llm, self.problem_text)

    async def run_workflow(self):
        """
        Implement the core problem-solving logic here.
        Remember:
        - Use detailed, comprehensive instructions
        - Dynamic instruction construction is powerful
        - All operators expect (instruction: str, context: str) except Ensemble which takes contexts: List[
        str]
        """
        import asyncio

        # Step 1: Extract key entities from the question
        extraction_instruction = (
        "Identify all named entities, key terms, and concepts in the question. "
        "Focus on entities that are likely to appear in the documents and are critical for answering the
        question."
        )
        question_entities = await self.generate(instruction=extraction_instruction, context=self.
        problem_text)

        # Step 2: First Vector Search to retrieve initial relevant documents
        first_search_instruction = (
        f"Use the extracted entities: {question_entities}. "
        "Perform a vector search to find documents that mention these entities. "
        "Prioritize documents that provide foundational information about the question."
        )
        first_retrieved_docs = await self.vector_search(instruction=first_search_instruction, context=self.
        problem_text, top_k=5)

        # Step 3: Analyze first retrieved documents to identify connections
```

```
        analysis_instruction = (
        f"Based on the retrieved documents: {first_retrieved_docs}. "
        "Identify any shared entities, relationships, or connections between the documents. "
        "Highlight how these connections can help answer the question."
        )
        connections_analysis = await self.generate(instruction=analysis_instruction, context=self.
        problem_text)

        # Step 4: Second Vector Search to retrieve documents connecting the entities
        second_search_instruction = (
        f"Use the connections identified: {connections_analysis}. "
        "Perform a vector search to find documents that explicitly connect the entities or provide
        additional context."
        )
        second_retrieved_docs = await self.vector_search(instruction=second_search_instruction, context=
        self.problem_text, top_k=5)

        # Step 5: Synthesize information from both document sets
        synthesis_instruction = (
        f"Combine the information from the first set: {first_retrieved_docs}, "
        f"and the second set: {second_retrieved_docs}. "
        "Extract the exact answer to the question based on the synthesized information. "
        "Ensure the answer is concise and factually accurate."
        )
        final_answer = await self.generate(instruction=synthesis_instruction, context=self.problem_text)

        return final_answer
```

Listing 14: Highest-scoring workflow for HotpotQA using the novel `VectorSearch` operator. The workflow performs iterative multi-hop reasoning: (1) entity extraction, (2) first-hop document retrieval, (3) connection analysis, (4) second-hop retrieval, and (5) answer synthesis. This demonstrates **MetaFlow**'s ability to compose complex operator sequences for unseen tools through learned meta-strategies.

ANALYSIS AND PERFORMANCE

the successful integration and application of a new tool. In Step 2, the model dynamically constructs a search query from its initial analysis and invokes the VECTORSEARCH operator, effectively performing active information retrieval. When evaluated on the HotpotQA downstream task, this approach achieved a **60% search accuracy**. This result is significant as it confirms that our operator framework enables models to learn and effectively utilize unseen tools.

NOTE ON COMPARABILITY WITH BASELINES

It is crucial to highlight a fundamental difference between our evaluation and that of many previous works on HotpotQA. Our methodology requires the model to **actively perform a search** to find relevant information. In contrast, prior baselines are often provided with the ground-truth supporting documents as part of their input, thereby bypassing the challenging information retrieval step entirely. Because our system solves a more complete and realistic version of the task that includes an explicit search phase, a direct comparison of end-to-end accuracy with such baselines is not meaningful.

