# OpenReview forum: "MetaFlow: A Meta Approach of Training LLMs into Generalizable Workflow Generators"
_ICLR.cc/2026/Conference — Submitted to ICLR 2026_

### Official Review · Reviewer_oAqH · 2025-10-29

**Soundness:** 1
**Presentation:** 1
**Contribution:** 1
**Rating:** 2
**Confidence:** 4

**Summary:**

This paper proposes MetaFlow, a meta-learning training method that enables LLMs to generate task-level workflows for unseen tasks and operators. MetaFlow includes two-stage training, where the first stage uses high-quality expert data (Qwen-Max) for supervised learning fine-tuning, and the second stage uses RLVR, with the former solving the cold-start problem of the latter. In the RLVR stage, the model updates its policy (SFT-finetuned Qwen3-8B) through the GRPO algorithm, using the average test score on the sampled data as the reward. Throught experiments on DROP, MBPP, GSM8K, MATH, MetaFlow demonstrates competitive performance on in-domain tasks.

**Strengths:**

- **Clear motivation**: Task-level approaches (ADAS, AFlow) require expensive re-optimization for new domains, while instance-level methods (ScoreFlow, FlowReasoner) generate per-query workflows without learning reusable patterns. MetaFlow addresses both via meta-learning for zero-shot generalization.
- **Reasonable writing structure**: The paper has clear structure with well-defined sections. The methodology (SFT + RLVR training, code-based workflow representation) is explained clearly.

**Weaknesses:**

- **Overclaimed novelty and contributions**: The paper uses standard SFT + GRPO, thus the only contribution is applying these to workflow generation. Just showing that if you train an LLM on selected tasks, it can generalize to similar tasks. This is expected behavior, not a novel insight.
- **Insufficient and unconvincing experiments**: The four questions in Section 5 are poorly addressed. (1) Table 1 formatting doesn't reflect best scores; (2) Missing ablation studies; (3) Claimed "OOD generalization" uses train/test splits from the same datasets (GSM8K, DROP, MBPP, HumanEval), not truly unseen domains; (4) Operator generalization shown only through case studies, lacking systematic statistical analysis.
- **Poor writing details**: Multiple presentation issues affect clarity. (1) Figure 1 has an uninformative caption and appears to be a low-quality raster image instead of a pdf or svg; (2) Code listings in the appendix lack syntax highlighting; (3) Critical details missing: learning rates, batch sizes, hardware, ...; (4) No code availability mentioned, harming reproducibility.

**Questions:**

- **Meta-learning justification**: MetaFlow trains on multiple tasks and tests on similar ones without gradient-based adaptation. How does this differ from standard multi-task learning? What specific meta-learning principle distinguishes MetaFlow?
- **Performance gap with ScoreFlow**: Table 1 shows MetaFlow (78.8) underperforms ScoreFlow (82.5). Why doesn't MetaFlow achieve at least comparable performance?

---

> ### Author Response · Authors · 2025-11-27
> **Author response to reviewer oAqH (Part 1)**
>
> We sincerely thank reviewer **oAqH** for the insightful feedback and constructive comments. Below, we address each point in detail.
>
> > W1. Overclaimed novelty and contributions: The paper uses standard SFT + GRPO, thus the only contribution is applying these to workflow generation. Just showing that if you train an LLM on selected tasks, it can generalize to similar tasks. This is expected behavior, not a novel insight.
>
> **A1**: We acknowledge this concern. Our contribution lies in constructing a structured pipeline specifically designed for online GRPO training in workflow generation, including API calling control and parallel execution of generated workflows across different instances. Additionally, we overcame significant obstacles such as model collapse during SFT, which is explained in $\texttt{Appendix B.2 (line 1119)}$.
>
> > W2-1. Table 1 formatting doesn't reflect best scores
>
> **A2-1**: In $\texttt{Section 5.4 (line 512)}$ of the updated version, we explain that MetaFlow and ScoreFlow represent two fundamentally different approaches. MetaFlow is a **cross-task generator** that produces a single workflow $\mathsf{W_C}$ per task $\mathsf{C}$. Consequently, any syntax error or incorrect operator call results in zero accuracy across all test instances. In contrast, ScoreFlow is an **instance-level approach** that generates a separate workflow $\mathsf{W_p}$ for each problem $\mathsf{p}$, meaning a failed generation affects only one data point. In terms of computational efficiency: if the validation set of task $\mathsf{C}$ contains $N$ problem instances, ScoreFlow requires $N$ inference calls while MetaFlow requires only a single inference. This fundamental paradigm difference explains the performance-efficiency trade-off.
>
> > W2-2. Missing ablation studies
>
> **A2-2**:
> + **Base model $\pi_\theta$ vs. $\pi_\theta^{\text{SFT}}$**: In $\texttt{Section 4.2.1 (line 293)}$ of the updated version, we explain that the SFT stage teaches the model to generate syntactically correct workflows following the required template structure. Without the SFT cold start, most of the generated workflows of the base model $\texttt{Qwen3-8B}$ do not math the template and thus cannot get executed.
> + **$\pi_\theta^{\text{SFT}}$ vs. $\pi_\theta^{\text{RL}}$**: The performance improvement can be observed from the training dynamics of the RL stage. The performance of the SFT checkpoint corresponds to the initial point (0.66) in Figure 3 (Right) at $\texttt{line 394}$.
>
> > W2-3. Claimed "OOD generalization" uses train/test splits from the same datasets (GSM8K, DROP, MBPP, HumanEval), not truly unseen domains;
>
> **A2-3**: Regarding "OOD generalization," we have added comprehensive experiments on the $\texttt{VectorSearch}$ operator in $\texttt{Section 5.3 (line 433)}$. This operator represents a genuinely novel capability involving tool calling rather than text processing (unlike $\texttt{Generate}$, $\texttt{Summarize}$, $\texttt{Revise}$, and $\texttt{Ensemble}$), providing stronger evidence for out-of-distribution generalization.
>
> > W2-4. Operator generalization shown only through case studies, lacking systematic statistical analysis.
>
> **A2-4**: Regarding systematic statistical analysis of operator generalization, we have added:
> + **Section 5.2**: Low-cost operator integration: We demonstrate that introducing a new operator requires only defining its interface, without model retraining or iterative search. These operators can be derived from manually-tested patterns such as self-consistency, which we abstract as the operator $\texttt{SelfConsistency}$.
> + **Section 5.3**: Zero-shot generalization to novel operators: We conducted rigorous testing on the $\texttt{VectorSearch}$ operator. We performed $100$ rollouts using the operator set $\mathsf{Ops} = \{\texttt{Generate}, \texttt{Summarize}, \texttt{Revise}, \texttt{Ensemble}, \texttt{VectorSearch}\}$ and evaluated the $100$ generated workflows on both the validation set ($100$ instances) and test set ($100$ instances). As shown in Figure 4, the sorted validation accuracy and corresponding test accuracy demonstrate clear improvement, growing from zero to surpass the baseline.

---

> ### Author Response · Authors · 2025-11-27
> **Author response to reviewer oAqH (Part 2)**
>
> > W3. Poor writing details: Multiple presentation issues affect clarity. (1) Figure 1 has an uninformative caption and appears to be a low-quality raster image instead of a pdf or svg; (2) Code listings in the appendix lack syntax highlighting; (3) Critical details missing: learning rates, batch sizes, hardware, ...; (4) No code availability mentioned, harming reproducibility.
>
> **A3**: We have significantly improved our figures and presentation quality. The new Figure 1 in $\texttt{Section 1 (line 57)}$ now illustrates the three different approaches with clear visual details, and the new Figure 2 in $\texttt{Section 4.1 (line 248)}$ presents the basic operators and the MetaFlow pipeline with enhanced clarity. We have also revised the appendix and added all implementation details: detailed operators design in $\texttt{Appendix A.1 (line 671)}$, input and output format in $\texttt{Appendix A.2 (line 876)}$, SFT dataset construction in $\texttt{Appendix B.1 (line 937)}$, SFT LoRA configurations in $\texttt{Appendix B.2 (line 1119)}$, and online RL configurations and environment in $\texttt{Appendix C (line 1134)}$.
>
> > Q1. Meta-learning justification: MetaFlow trains on multiple tasks and tests on similar ones without gradient-based adaptation. How does this differ from standard multi-task learning? What specific meta-learning principle distinguishes MetaFlow?
>
> **A4**: We clarify that MetaFlow does employ gradient-based adaptation in both the SFT and RL stages. Regarding "meta-learning," our approach is grounded in the principle that by training the base model on diverse (task, operator set) pairs $(\mathsf{C}, \mathsf{Ops})$, we enable generalization to both novel tasks and novel operator sets. This meta-level capability allows the model to learn how to compose workflows rather than memorizing task-specific solutions. We have added the following evidence to support this claim (see **Section 5.2** and **Section 5.3** detailed in **A2-4** above).
>
> > Q2. Performance gap with ScoreFlow: Table 1 shows MetaFlow (78.8) underperforms ScoreFlow (82.5). Why doesn't MetaFlow achieve at least comparable performance?
>
> **A5**: In $\texttt{Section 5.4 (line 512)}$ of the updated version, we explain that MetaFlow and ScoreFlow represent two fundamentally different approaches. MetaFlow is a **cross-task generator** that produces a single workflow $\mathsf{W_C}$ per task $\mathsf{C}$. Consequently, any syntax error or incorrect operator call results in zero accuracy across all test instances. In contrast, ScoreFlow is an **instance-level approach** that generates a separate workflow $\mathsf{W_p}$ for each problem $\mathsf{p}$, meaning a failed generation affects only one data point. In terms of computational efficiency: if the validation set of task $\mathsf{C}$ contains $N$ problem instances, ScoreFlow requires $N$ inference calls while MetaFlow requires only a single inference. This fundamental paradigm difference explains the performance-efficiency trade-off.

---

> > ### Comment · Reviewer_oAqH · 2025-11-27
> >
> > Thanks for author's detailed response.
> >
> > Most of my concerns have been resolved. However, it must be pointed out that the bolded items in Table 1 must be replaced, which is commonly used to indicate the optimal score in the table. If you want to differentiate the performance of MetaFlow, it is recommended to use a different method, such as adding a background color to the row.

---

> > > ### Author Response · Authors · 2025-11-27
> > > **Author response to reviewer oAqH**
> > >
> > > We sincerely thank Reviewer **oAqH** for the quick and thoughtful feedback. We have revised the table as suggested (in the newly updated PDF)—the bold formatting has been removed, and we now use a subtle background color to distinguish MetaFlow's results. This approach aligns better with standard conventions while maintaining clarity.
> > >
> > > With all major concerns now addressed, we believe these revisions have strengthened the paper and improved its overall quality. We would be grateful if you could reconsider the rating in light of these improvements.
> > >
> > > Thank you again for your time and constructive suggestions throughout the review process.

---

> > > > ### Comment · Reviewer_oAqH · 2025-11-27
> > > >
> > > > Thank you for the author's positive response. I have no more questions. Considering that this paper has supplemented sufficient content and revised the key issues, I will raise the score to 4 points.

---

### Official Review · Reviewer_qGS7 · 2025-10-30

**Soundness:** 1
**Presentation:** 1
**Contribution:** 2
**Rating:** 2
**Confidence:** 5

**Summary:**

This paper proposes MetaFlow, reformulating workflow generation as a meta-learning problem. Given a task description C and operator set Ops, the model learns to directly synthesize workflows W through single inference, enabling zero-shot generation. MetaFlow employs two-stage training: (1) supervised fine-tuning on 1,300 synthetic workflows using Qwen3-8B; (2) reinforcement learning with verifiable rewards (RLVR) using the GRPO algorithm, training for 137 steps across 4 tasks to optimize expected rewards over the task-operator distribution. Experiments show MetaFlow achieves an average score of 78.8 on in-domain tasks (compared to ScoreFlow's 82.5), and demonstrates zero-shot generalization to the Programmer operator and MATH task.

**Strengths:**

1. **Clear Problem Formulation**: The paper identifies two fundamental limitations of existing approaches—task-level methods require expensive re-optimization for each new task, while instance-level methods cannot reuse task-level patterns—and proposes meta-learning as an alternative path. The motivation is clearly articulated and the problem setting is well-justified.

2. **Rigorous Formalization**: Section 3 formulates workflow generation as a bi-level optimization problem, explicitly defining the task distribution, fast adaptation mechanism, and meta-optimization objective, establishing clear connections to classic meta-learning literature (MAML, Franceschi et al. 2018).

3. **Informative Qualitative Cases**: The appendix's before-and-after training comparison demonstrates workflow design improvements (e.g., replacing information compression with structured extraction, parallel candidate generation), which, though qualitative, helps understand training effects. Appendix A showcases learning to use the VectorSearch operator on HotpotQA.

**Weaknesses:**

**1. Insufficient Evidence for Core Claims**

The paper's core value proposition is "zero-shot generalization to unseen tasks and operators," but critical baselines and details are missing:

- **Missing Base Model Baselines**: No performance reported for Qwen3-8B zero-shot/few-shot workflow generation, making it impossible to verify whether the 78.8 score stems from meta-learning training or the base LLM's in-context learning ability. Given complete natural language operator descriptions, the base LLM might achieve similar results.

- **MATH is Not Truly OOD**: Even if SFT excludes MATH samples, MATH cannot be considered truly out-of-distribution. The planner uses Qwen3-8B, which has seen extensive mathematical data during pretraining and possesses strong mathematical capabilities matching MATH dataset difficulty.

- **Novel Operator Generalization Not Proven from Training**: The paper claims MetaFlow can use the unseen Programmer operator during training (Section 5.2.2). While Appendix A.0.1 provides qualitative before-and-after cases, a single example cannot rule out performance differences due to sampling randomness. Quantitative performance experiments are still missing: no reporting of accuracy differences between "trained MetaFlow vs untrained Qwen3-8B" when using the Programmer operator. Given complete natural language operator descriptions, Qwen3-8B itself may already possess the ability to use new operators. Quantitative baselines of the base model across all test tasks are needed to prove this generalization capability comes from meta-learning training rather than the model's inherent instruction-following ability.

**2. Cannot Separate SFT vs RLVR Contributions**

The paper claims the "meta-learning framework" is the core contribution, but lacks a critical ablation: no performance reported for "SFT only (without RLVR)". This is a fundamental attribution problem—unable to quantify RLVR's incremental contribution over SFT, or verify whether zero-shot generalization comes from cross-task meta-optimization or simply supervised learning on 1,300 diverse samples.

**3. Performance Consistently Inferior to ScoreFlow Without Explanation**

MetaFlow (78.8) underperforms ScoreFlow (82.5) across all tasks: DROP (-3.4), MBPP (-7.2), GSM8K (-0.8), MATH (-3.4). Theoretically, meta-learning should achieve stronger generalization through cross-task learning, at least matching on in-domain tasks. The paper claims "comparable" performance to SOTA, but Table 1 shows consistent underperformance. Possible explanations include: insufficient Qwen3-8B capacity, SFT data quality issues, inadequate RLVR training (only 137 steps), or single-task methods' deep optimization advantages—none are discussed. The largest MBPP gap (-7.2) particularly needs explanation, along with whether the performance-generalization tradeoff is acceptable.

**4. Incomplete and Inconsistent Experimental Details**

- **RL Step Count Inconsistency**: Text states GRPO runs for 137 steps, but figures only show up to 100 iterations, with no explanation.

- **Opaque SFT Data**: 1,300 samples generated by Qwen-Max, covering "four tasks" and "a single operator set," but does not specify which four tasks, the Qwen-Max prompts, or how sample quality was verified (human review?). If SFT includes MATH, zero-shot claims fail; if quality is poor, this may explain underperformance vs ScoreFlow.

- **Missing Cost/Latency Reports**: No specific token consumption and runtime provided for training (SFT data creation, RLVR iterations) and inference (candidate generation, validation selection, execution), or comparison with single-task search methods. GRPO requires executing k candidate workflows on N instances per step to compute rewards, but k, N, and average operator calls per workflow-instance pair are unreported.

- **best-of-20 Strategy Consistency**: Paper claims "single inference" avoids search, but evaluation generates 20 candidates and selects the best on 50 validation instances, which is essentially search (though lighter than AFlow's 20-round iterations).


**5. Figure Quality Issues**

The paper's schematic diagrams (e.g., Figure 1, Figure 5) suffer from readability problems: text in figures is too small and blurry, making it difficult to discern key labels and process descriptions. In contrast, statistical plots (e.g., training curves in Figures 2-4) use excessively large fonts that occupy too much space. Additionally, the figure organization is illogical: Figure 2 (SFT loss curve) and Figure 3 (RL running reward) are actually subfigures of Figure 4 (Training curves) and should not be presented as separate peer-level figures, leading to confusing numbering and wasted space. The figure design should be revised to ensure schematic text is clear and legible, statistical plot fonts are appropriately sized, and figure hierarchy is properly organized to improve overall presentation quality.

**Questions:**

1. **Missing Baselines and Controls**:
   - Please provide zero-shot/few-shot workflow generation baselines for the untrained base model (Qwen3-8B) across all evaluation tasks, to verify whether the 78.8 improvement stems from meta-learning or the base LLM's instruction-following ability.
   - Please provide performance for "SFT only (without RLVR)" and sensitivity analysis across different RL training steps, to quantify RLVR's incremental contribution. Text states GRPO trains for 137 steps while figures show only to 100 iterations—please explain this inconsistency and provide convergence justification.
   - For unseen operators (e.g., Programmer), please report quantitative comparisons of "trained Planner vs untrained Qwen3-8B" to support the core claim of "zero-shot novel operator integration." The qualitative case in the appendix is insufficient to replace systematic comparison.

2. **SFT Data and Transparency**:
   - Clarify the specific composition of the "four tasks" in the SFT stage, whether they align with the RL stage's GSM8K/DROP/MBPP/HumanEval, and their respective sample proportions; specify whether MATH-related samples are included.
   - Provide the prompt template/procedure for Qwen-Max sample generation, and the quality verification method for "1,300 high-quality samples" (human review? selection criteria?). If possible, please release a subset of the SFT data for verification.

3. **Evaluation Protocol and Terminology**:
   - The paper claims "single inference," but evaluation uses best-of-20 (selecting the best candidate on 50 validation instances). Please clarify the relationship between this procedure and "single inference."
   - For fair comparison, please specify whether all methods uniformly use the same candidate selection and validation protocol; and report specific inference costs (candidate count, validation set size, execution call volume).

4. **Performance Gap Analysis**:
   - Table 1 shows MetaFlow underperforms ScoreFlow across all tasks (especially the large MBPP gap). Please provide possible causes and analysis (e.g., planner scale, SFT data distribution, RLVR training budget limitations), and discuss the performance-generalization tradeoff.

---

> ### Author Response · Authors · 2025-11-27
> **Author response to reviewer qGS7 (Part 1)**
>
> We sincerely thank reviewer **qGS7** for the insightful feedback and constructive comments. Below, we address each point in detail.
>
> > W1-1. Missing Base Model Baselines: No performance reported for Qwen3-8B zero-shot/few-shot workflow generation, making it impossible to verify whether the 78.8 score stems from meta-learning training or the base LLM's in-context learning ability. Given complete natural language operator descriptions, the base LLM might achieve similar results.
>
> **A1-1**:
> + **Base model $\pi_\theta$ vs. $\pi_\theta^{\text{SFT}}$**: In $\texttt{Section 4.2.1 (line 293)}$ of the updated version, we explain that the SFT stage teaches the model to generate syntactically correct workflows following the required template structure. Without the SFT cold start, most of the generated workflows of the base model $\texttt{Qwen3-8B}$ do not math the template and thus cannot get executed.
> + **$\pi_\theta^{\text{SFT}}$ vs. $\pi_\theta^{\text{RL}}$**: The performance improvement can be observed from the training dynamics of the RL stage. The performance of the SFT checkpoint corresponds to the initial point (0.66) in Figure 3 (Right) at $\texttt{line 394}$.
>
> > W1-2. MATH is Not Truly OOD: Even if SFT excludes MATH samples, MATH cannot be considered truly out-of-distribution. The planner uses Qwen3-8B, which has seen extensive mathematical data during pretraining and possesses strong mathematical capabilities matching MATH dataset difficulty.
>
> **A1-2**: We acknowledge this concern. Regarding "OOD generalization," we have added comprehensive experiments on the $\texttt{VectorSearch}$ operator in $\texttt{Section 5.3 (line 433)}$. This operator represents a genuinely novel capability involving tool calling rather than text processing (unlike $\texttt{Generate}$, $\texttt{Summarize}$, $\texttt{Revise}$, and $\texttt{Ensemble}$), providing stronger evidence for out-of-distribution generalization.
>
> > W1-3. Novel Operator Generalization Not Proven from Training: The paper claims MetaFlow can use the unseen Programmer operator during training (Section 5.2.2). While Appendix A.0.1 provides qualitative before-and-after cases, a single example cannot rule out performance differences due to sampling randomness. Quantitative performance experiments are still missing: no reporting of accuracy differences between "trained MetaFlow vs untrained Qwen3-8B" when using the Programmer operator. Given complete natural language operator descriptions, Qwen3-8B itself may already possess the ability to use new operators. Quantitative baselines of the base model across all test tasks are needed to prove this generalization capability comes from meta-learning training rather than the model's inherent instruction-following ability.
>
> **A1-3**: In $\texttt{Section 4.2.1 (line 293)}$ of the updated version, we explain that the SFT stage teaches the model to generate syntactically correct workflows following the required template structure (this also contains the fixed operator calling method). the SFT cold start, most of the generated workflows of the base model $\texttt{Qwen3-8B}$ do not math the template or the calling of the operator in these workflows is not right and thus cannot get executed.
>
> > W2. Cannot Separate SFT vs RLVR Contributions
>
> **A2**: We acknowledge this concern. Our contribution lies in constructing a structured pipeline specifically designed for online GRPO training in workflow generation, including API calling control and parallel execution of generated workflows across different instances. Additionally, we overcame significant obstacles such as model collapse during SFT, which is explained in $\texttt{Appendix B.2 (line 1119)}$.
>
> > W3. Performance Consistently Inferior to ScoreFlow Without Explanation
>
> **A3**: In $\texttt{Section 5.4 (line 512)}$ of the updated version, we explain that MetaFlow and ScoreFlow represent two fundamentally different approaches. MetaFlow is a **cross-task generator** that produces a single workflow $\mathsf{W_C}$ per task $\mathsf{C}$. Consequently, any syntax error or incorrect operator call results in zero accuracy across all test instances. In contrast, ScoreFlow is an **instance-level approach** that generates a separate workflow $\mathsf{W_p}$ for each problem $\mathsf{p}$, meaning a failed generation affects only one data point. In terms of computational efficiency: if the validation set of task $\mathsf{C}$ contains $N$ problem instances, ScoreFlow requires $N$ inference calls while MetaFlow requires only a single inference. This fundamental paradigm difference explains the performance-efficiency trade-off.

---

> ### Author Response · Authors · 2025-11-27
> **Author response to reviewer qGS7 (Part 2)**
>
> > W4-1. RL Step Count Inconsistency: Text states GRPO runs for 137 steps, but figures only show up to 100 iterations, with no explanation.
>
> **A4-1**: In **Section 5.1**, we explain that the total training steps is $137$ and we choose the checkpoint at the $100$ steps for evaluation. We have also updated the figure of the RL training dynamics.
>
> > W4-2. Opaque SFT Data: 1,300 samples generated by Qwen-Max, covering "four tasks" and "a single operator set," but does not specify which four tasks, the Qwen-Max prompts, or how sample quality was verified (human review?). If SFT includes MATH, zero-shot claims fail; if quality is poor, this may explain underperformance vs ScoreFlow.
>
> **A4-2**: We add the details to the updated version of the paper at **Section 5.1** with detailed operators design in $\texttt{Appendix A.1 (line 671)}$, input and output format in $\texttt{Appendix A.2 (line 876)}$, SFT dataset construction in $\texttt{Appendix B.1 (line 937)}$, SFT LoRA configurations in $\texttt{Appendix B.2 (line 1119)}$, and online RL configurations and environment in $\texttt{Appendix C (line 1134)}$.
>
> > W4-3. Missing Cost/Latency Reports: No specific token consumption and runtime provided for training (SFT data creation, RLVR iterations) and inference (candidate generation, validation selection, execution), or comparison with single-task search methods. GRPO requires executing k candidate workflows on N instances per step to compute rewards, but k, N, and average operator calls per workflow-instance pair are unreported.
>
> **A4-3**:
> + SFT data creation: SFT dataset construction in $\texttt{Appendix B.1 (line 937)}$.
> + RLVR iterations: $137$ steps in total and we choose the checkpoint at $100$ steps for evaluation.
> + Inference (candidate generation, validation selection, execution): operators design in $\texttt{Appendix A.1 (line 671)}$, input and output format in $\texttt{Appendix A.2 (line 876)}$.
> + GRPO configurations: rollout is $8$ and batch size of each step is $24$.
>
> > W4-4. best-of-20 Strategy Consistency: Paper claims "single inference" avoids search, but evaluation generates 20 candidates and selects the best on 50 validation instances, which is essentially search (though lighter than AFlow's 20-round iterations).
>
> **A4-4**: We have added experiments on the $\texttt{VectorSearch}$ operator in $\texttt{Section 5.3 (line 433)}$. Here we make $100$ rollout to get $100$ workflows and we execute each of the workflow on validation set ($100$ instances) and test set ($100$ instances). We can figure out that, the best performance on validation set and test set both get better with more rollout. We acknowledge that this search method's cost (mainly on calling the api for execution of the workflow) is larger than just single inference. But the cost is lighter than iteratively searching method like AFlow and our searching method is totally parallel (the generation of each workflow and the execution of the workflow on each problem instance) and thus more scalable with the meta-ability (using new operators without training).

---

> ### Author Response · Authors · 2025-11-27
> **Author response to reviewer qGS7 (Part 3)**
>
> > W5. Figure Quality Issues
>
> **A5**: We have significantly improved our figures and presentation quality. The new Figure 1 in $\texttt{Section 1 (line 57)}$ now illustrates the three different approaches with clear visual details, and the new Figure 2 in $\texttt{Section 4.1 (line 248)}$ presents the basic operators and the MetaFlow pipeline with enhanced clarity.
>
> > Q1-1. Please provide zero-shot/few-shot workflow generation baselines for the untrained base model (Qwen3-8B) across all evaluation tasks, to verify whether the 78.8 improvement stems from meta-learning or the base LLM's instruction-following ability.
>
> **A6-1**:  In $\texttt{Section 4.2.1 (line 293)}$ of the updated version, we explain that the SFT stage teaches the model to generate syntactically correct workflows following the required template structure (this also contains the fixed operator calling method). the SFT cold start, most of the generated workflows of the base model $\texttt{Qwen3-8B}$ do not match the template or the calling of the operator in these workflows is not right and thus cannot get executed.
>
> > Q1-2. Please provide performance for "SFT only (without RLVR)" and sensitivity analysis across different RL training steps, to quantify RLVR's incremental contribution. Text states GRPO trains for 137 steps while figures show only to 100 iterations—please explain this inconsistency and provide convergence justification.
>
> **A6-2**: **$\pi_\theta^{\text{SFT}}$ vs. $\pi_\theta^{\text{RL}}$**: The performance improvement can be observed from the training dynamics of the RL stage. The performance of the SFT checkpoint corresponds to the initial point (0.66) in Figure 3 (Right) at $\texttt{line 394}$.
>
> > Q1-3. For unseen operators (e.g., Programmer), please report quantitative comparisons of "trained Planner vs untrained Qwen3-8B" to support the core claim of "zero-shot novel operator integration." The qualitative case in the appendix is insufficient to replace systematic comparison.
>
> **A6-3**: For the untrained Qwen3-8B, this model can not generate workflows that match the template or the calling of the operator under the same prompt to the trained $\pi_\theta^{\text{RL}}$. Indeed, if making the same $100$ rollout on Qwen3-8B in **Section 5.3**, the generated $100$ workflows cannot get executed.
>
> > Q2. SFT Data and Transparency:
>
> **A7**: We provide detailed operators design in $\texttt{Appendix A.1 (line 671)}$, input and output format in $\texttt{Appendix A.2 (line 876)}$, SFT dataset construction in $\texttt{Appendix B.1 (line 937)}$, SFT LoRA configurations in $\texttt{Appendix B.2 (line 1119)}$, and online RL configurations and environment in $\texttt{Appendix C (line 1134)}$.

---

> ### Author Response · Authors · 2025-11-27
> **Author response to reviewer qGS7 (Part 4)**
>
> > Q3. Evaluation Protocol and Terminology
>
> **A8**:
> + **Paradigm Differences**: In $\texttt{Section 5.4 (line 512)}$ of the updated version, we explain that MetaFlow and ScoreFlow represent two fundamentally different approaches. MetaFlow is a **cross-task generator** that produces a single workflow $\mathsf{W_C}$ per task $\mathsf{C}$. Consequently, any syntax error or incorrect operator call results in zero accuracy across all test instances. In contrast, ScoreFlow is an **instance-level approach** that generates a separate workflow $\mathsf{W_p}$ for each problem $\mathsf{p}$, meaning a failed generation affects only one data point. In terms of computational efficiency: if the validation set of task $\mathsf{C}$ contains $N$ problem instances, ScoreFlow requires $N$ inference calls while MetaFlow requires only a single inference. This fundamental paradigm difference explains the performance-efficiency trade-off.
> + **best-of-20**:  We have added experiments on the $\texttt{VectorSearch}$ operator in $\texttt{Section 5.3 (line 433)}$. Here we make $100$ rollout to get $100$ workflows and we execute each of the workflow on validation set ($100$ instances) and test set ($100$ instances). We can figure out that, the best performance on validation set and test set both get better with more rollout. We acknowledge that this search method's cost (mainly on calling the api for execution of the workflow) is larger than just single inference. But the cost is lighter than iteratively searching method like AFlow and our searching method is totally parallel (the generation of each workflow and the execution of the workflow on each problem instance) and thus more scalable with the meta-ability (using new operators without training).
>
> > Q4. Performance Gap Analysis
>
> **A9**:  In $\texttt{Section 5.4 (line 512)}$ of the updated version, we explain that MetaFlow and ScoreFlow represent two fundamentally different approaches. MetaFlow is a **cross-task generator** that produces a single workflow $\mathsf{W_C}$ per task $\mathsf{C}$. Consequently, any syntax error or incorrect operator call results in zero accuracy across all test instances. In contrast, ScoreFlow is an **instance-level approach** that generates a separate workflow $\mathsf{W_p}$ for each problem $\mathsf{p}$, meaning a failed generation affects only one data point. In terms of computational efficiency: if the validation set of task $\mathsf{C}$ contains $N$ problem instances, ScoreFlow requires $N$ inference calls while MetaFlow requires only a single inference. This fundamental paradigm difference explains the performance-efficiency trade-off.

---

### Official Review · Reviewer_2R4k · 2025-10-31

**Soundness:** 3
**Presentation:** 1
**Contribution:** 2
**Rating:** 4
**Confidence:** 5

**Summary:**

This paper proposes MetaFlow, a meta-learning framework for generalizable workflow generation in LLM-based agents. Instead of generating task-specific or instance-level workflows, MetaFlow learns a meta-policy that maps task and operator descriptions to reusable workflows through a two-stage training pipeline (SFT + RL with verifiable rewards). The paper aims to improve generalization across tasks and operators, pushing toward scalable automation of agentic workflow generation.

**Strengths:**

1. The paper tackles an important problem: generalizing workflow generation beyond specific tasks or instances, which is timely and relevant for the automation of LLM-based agents.
2. The meta-learning formulation is conceptually elegant and represents a clear shift from instance-level reasoning (as in FlowReasoner) to task-level generalization.
3. The qualitative analysis and case studies (e.g., learning unseen operators like VectorSearch) provide some interpretability and insight into the learned workflows.

**Weaknesses:**

1. Experimental results are weak: MetaFlow performs worse than several baselines (including AFlow and ScoreFlow) and fails to reach SOTA. Moreover, the omission of FlowReasoner, a direct conceptual baseline, makes the evaluation incomplete.

2. Figures are poorly designed — Figure 1 and Figure 5 use raster graphics, and Figure 4’s oversized fonts distort layout — which undermines the professionalism expected of a top-tier paper.

3. Successfully using a new operator does not convincingly demonstrate true out-of-distribution generalization. The authors need to give more evidence instead of only use the datasets used by aflow and scoreflow.

**Questions:**

The paper’s main conceptual gap lies in its unclear definition of “meta-learning” and “OOD generalization.” The method essentially performs conditional text generation rather than true meta-level adaptation, and the evidence provided (e.g., new operator usage) does not convincingly demonstrate structural generalization. So the key question is the evidence, authors need more evidence to fill their claim.

---

> ### Author Response · Authors · 2025-11-27
> **Author response to reviewer 2R4k**
>
> We sincerely thank reviewer **2R4k** for the insightful feedback and constructive comments. Below, we address each point in detail.
>
> > W1. Experimental results are weak: MetaFlow performs worse than several baselines (including AFlow and ScoreFlow) and fails to reach SOTA. Moreover, the omission of FlowReasoner, a direct conceptual baseline, makes the evaluation incomplete.
>
> **A1**: In $\texttt{Section 5.4 (line 512)}$ of the updated version, we explain that MetaFlow and ScoreFlow represent two fundamentally different approaches. MetaFlow is a **cross-task generator** that produces a single workflow $\mathsf{W_C}$ per task $\mathsf{C}$. Consequently, any syntax error or incorrect operator call results in zero accuracy across all test instances. In contrast, ScoreFlow is an **instance-level approach** that generates a separate workflow $\mathsf{W_p}$ for each problem $\mathsf{p}$, meaning a failed generation affects only one data point. In terms of computational efficiency: if the validation set of task $\mathsf{C}$ contains $N$ problem instances, ScoreFlow requires $N$ inference calls while MetaFlow requires only a single inference. This fundamental paradigm difference explains the performance-efficiency trade-off.
>
> Regarding FlowReasoner, we initially omitted it because it follows the same **instance-level** approach as ScoreFlow and primarily focuses on coding tasks. We have now added FlowReasoner to the comparison in the updated version of the paper. (see $\texttt{line 52}$ and $\texttt{line 491}$)
>
> > W2. Figures are poorly designed — Figure 1 and Figure 5 use raster graphics, and Figure 4's oversized fonts distort layout — which undermines the professionalism expected of a top-tier paper.
>
> **A2**: We have significantly improved our figures and presentation quality. The new Figure 1 in $\texttt{Section 1 (line 57)}$ now illustrates the three different approaches with clear visual details, and the new Figure 2 in $\texttt{Section 4.1 (line 248)}$ presents the basic operators and the MetaFlow pipeline with enhanced clarity.
>
> > W3. Successfully using a new operator does not convincingly demonstrate true out-of-distribution generalization. The authors need to give more evidence instead of only use the datasets used by AFlow and ScoreFlow.
>
> **A3**: Regarding "OOD generalization," we have added comprehensive experiments on the $\texttt{VectorSearch}$ operator in $\texttt{Section 5.3 (line 433)}$. This operator represents a genuinely novel capability involving tool calling rather than text processing (unlike $\texttt{Generate}$, $\texttt{Summarize}$, $\texttt{Revise}$, and $\texttt{Ensemble}$), providing stronger evidence for out-of-distribution generalization.
>
> > Q1. The paper's main conceptual gap lies in its unclear definition of "meta-learning" and "OOD generalization." The method essentially performs conditional text generation rather than true meta-level adaptation, and the evidence provided (e.g., new operator usage) does not convincingly demonstrate structural generalization. So the key question is the evidence, authors need more evidence to fill their claim.
>
> **A4**: Regarding "meta-learning," our approach is grounded in the principle that by training the base model on diverse (task, operator set) pairs $(\mathsf{C}, \mathsf{Ops})$, we enable generalization to both novel tasks and novel operator sets. This meta-level capability allows the model to learn how to compose workflows rather than memorizing task-specific solutions. We have added the following evidence to support this claim:
> + **Section 5.2**: Low-cost operator integration: We demonstrate that introducing a new operator requires only defining its interface, without model retraining or iterative search. These operators can be derived from manually-tested patterns such as self-consistency, which we abstract as the operator $\texttt{SelfConsistency}$. This shows the model has learned composable workflow generation rather than pattern matching.
> + **Section 5.3**: Zero-shot generalization to novel operators: We conducted rigorous testing on the $\texttt{VectorSearch}$ operator. We performed $100$ rollouts using the operator set $\mathsf{Ops} = \{\texttt{Generate}, \texttt{Summarize}, \texttt{Revise}, \texttt{Ensemble}, \texttt{VectorSearch}\}$ and evaluated the $100$ generated workflows on both the validation set ($100$ instances) and test set ($100$ instances). As shown in Figure 4, the sorted validation accuracy and corresponding test accuracy demonstrate clear improvement, growing from zero to surpass the baseline. This structural generalization to a tool-calling operator (fundamentally different from text-processing operators) provides strong evidence of meta-level adaptation.

---

### Official Review · Reviewer_yP2t · 2025-11-01

**Soundness:** 3
**Presentation:** 3
**Contribution:** 3
**Rating:** 2
**Confidence:** 5

**Summary:**

This paper presents MetaFlow, a meta-learning framework that reformulates workflow generation from task-specific optimization to learning generalizable workflow construction strategies across diverse task-operator combinations. Through a two-stage training paradigm combining supervised fine-tuning and reinforcement learning with verifiable rewards (RLVR), MetaFlow learns to synthesize workflows via a single inference without requiring re-optimization for new domains. While the approach offers a novel perspective on automatic workflow generation, the experimental results demonstrate limited improvements over existing methods, and the presentation and experimental analysis require substantial strengthening.

**Strengths:**

The paper introduces a novel meta-learning perspective to workflow generation that shifts from instance-level or task-specific optimization to learning cross-domain generalization capabilities, which is conceptually interesting and worth encouraging despite suboptimal performance.

**Weaknesses:**

W1. The experimental results are underwhelming, with MetaFlow achieving an average score of 78.8 compared to ScoreFlow's 82.5, showing almost no state-of-the-art performance on any benchmark and raising questions about the practical value of the proposed approach.

W2. The presentation quality needs significant improvement: Figure 1 merely lists text without visual details (consider FlowReasoner's illustration style); Figure 5's operator visualization appears unnecessary; and the overall figure quality lacks the polish expected for a top-tier venue.

W3. The experimental section is inadequately developed, dedicating excessive space to configuration details while providing minimal analysis, ablation studies, or cost comparisons, which undermines confidence in the thoroughness of the empirical validation.

W4. Several claims are overstated or insufficiently supported: the "OOD generalization" in Section 5.2.2 is questionable since MATH shares characteristics with GSM8K (both mathematical reasoning), and the novel operator integration (Decompose, Programmer) represents a common capability in prior automatic workflow literature rather than a unique contribution.

**Questions:**

Could you provide more detailed ablation studies showing the individual contributions of SFT vs. RLVR stages, and include a comprehensive cost analysis comparing the computational overhead of your single-inference approach against iterative methods like ScoreFlow and AFlow across different scales?

---

> ### Author Response · Authors · 2025-11-27
> **Author response to reviewer yP2t**
>
> We sincerely thank reviewer **yP2t** for the insightful feedback and constructive comments. Below, we address each point in detail.
>
> > W1. The experimental results are underwhelming, with MetaFlow achieving an average score of 78.8 compared to ScoreFlow's 82.5, showing almost no state-of-the-art performance on any benchmark and raising questions about the practical value of the proposed approach.
>
> **A1**: In $\texttt{Section 5.4 (line 512)}$ of the updated version, we explain that MetaFlow and ScoreFlow represent two fundamentally different approaches. MetaFlow is a **cross-task generator** that produces a single workflow $\mathsf{W_C}$ per task $\mathsf{C}$. Consequently, any syntax error or incorrect operator call results in zero accuracy across all test instances. In contrast, ScoreFlow is an **instance-level approach** that generates a separate workflow $\mathsf{W_p}$ for each problem $\mathsf{p}$, meaning a failed generation affects only one data point. In terms of computational efficiency: if the validation set of task $\mathsf{C}$ contains $N$ problem instances, ScoreFlow requires $N$ inference calls while MetaFlow requires only a single inference. This fundamental paradigm difference explains the performance-efficiency trade-off.
>
> > W2. The presentation quality needs significant improvement: Figure 1 merely lists text without visual details (consider FlowReasoner's illustration style); Figure 5's operator visualization appears unnecessary; and the overall figure quality lacks the polish expected for a top-tier venue.
>
> **A2**: We have significantly improved our figures and presentation quality. The new Figure 1 in $\texttt{Section 1 (line 57)}$ now illustrates the three different approaches with clear visual details, and the new Figure 2 in $\texttt{Section 4.1 (line 248)}$ presents the basic operators and the MetaFlow pipeline with enhanced clarity.
>
> > W3. The experimental section is inadequately developed, dedicating excessive space to configuration details while providing minimal analysis, ablation studies, or cost comparisons, which undermines confidence in the thoroughness of the empirical validation.
>
> **A3**: We have substantially revised the experimental section and added comprehensive ablation studies:
> + **Section 5.1**: Training configurations, with detailed operators design in $\texttt{Appendix A.1 (line 671)}$, input and output format in $\texttt{Appendix A.2 (line 876)}$, SFT dataset construction in $\texttt{Appendix B.1 (line 937)}$, SFT LoRA configurations in $\texttt{Appendix B.2 (line 1119)}$, and online RL configurations and environment in $\texttt{Appendix C (line 1134)}$.
> + **Section 5.2**: Low-cost operator integration: We demonstrate that introducing a new operator requires only defining its interface, without model retraining or iterative search. These operators can be derived from manually-tested patterns such as self-consistency, which we abstract as the operator $\texttt{SelfConsistency}$.
> + **Section 5.3**: Zero-shot generalization to novel operators: We conducted rigorous testing on the $\texttt{VectorSearch}$ operator. We performed $100$ rollouts using the operator set $\mathsf{Ops} = \{\texttt{Generate}, \texttt{Summarize}, \texttt{Revise}, \texttt{Ensemble}, \texttt{VectorSearch}\}$ and evaluated the $100$ generated workflows on both the validation set ($100$ instances) and test set ($100$ instances). As shown in Figure 4, the sorted validation accuracy and corresponding test accuracy demonstrate clear improvement, growing from zero to surpass the baseline.
> + **Section 5.4**: Main results and analysis: We added a detailed explanation of paradigm differences and trade-offs at $\texttt{line 509}$. Specifically, MetaFlow (cross-task generator) produces one workflow $\mathsf{W_C}$ per task $\mathsf{C}$ with single inference, while ScoreFlow (instance-level approach) generates $\mathsf{W_p}$ per problem $\mathsf{p}$ with $N$ inferences for $N$ instances, which explains the performance-efficiency trade-off observed in the results.
>
> > W4. Several claims are overstated or insufficiently supported: the "OOD generalization" in Section 5.2.2 is questionable since MATH shares characteristics with GSM8K (both mathematical reasoning), and the novel operator integration (Decompose, Programmer) represents a common capability in prior automatic workflow literature rather than a unique contribution.
>
> **A4**: Regarding "OOD generalization," we have added comprehensive experiments on the $\texttt{VectorSearch}$ operator in $\texttt{Section 5.3 (line 433)}$. This operator represents a genuinely novel capability involving tool calling rather than text processing (unlike $\texttt{Generate}$, $\texttt{Summarize}$, $\texttt{Revise}$, and $\texttt{Ensemble}$), providing stronger evidence for out-of-distribution generalization.

---

### Author Response · Authors · 2025-12-04
**Global Response (Part 1)**

We sincerely thank the reviewers for their detailed and constructive feedback.

In this round of response, we have made the following revisions.

### 1. Clarifying the Fundamental Paradigm: Cross-Task vs. Instance-Level Generation
**Addressed concerns from:**
*   **Reviewer yP2t (W1):** Questioned the "underwhelming" performance (78.8 vs ScoreFlow's 82.5) and practical value.
*   **Reviewer 2R4k (W1):** Noted failure to reach SOTA and lower performance than baselines like AFlow.
*   **Reviewer qGS7 (W3, Q4):** Pointed out consistent performance gaps and requested explanation for the trade-off.
*   **Reviewer oAqH (W2-1, Q2):** Asked why MetaFlow does not achieve comparable performance to ScoreFlow.

**Our Response and Revision:**
We realized that our original manuscript failed to clearly articulate the fundamental structural difference between MetaFlow and the baselines. We have added a dedicated theoretical discussion in **$\texttt{Section 5.4 (line 512)}$** to explicitly define this trade-off using formal notation.

*   **MetaFlow is a Cross-Task Generator:** We define our approach as generating a single, universal workflow $\mathsf{W_C}$ for a specific task category $\mathsf{C}$ (e.g., "Solve math word problems"). Once generated, this single workflow template is applied to *all* test instances.
    *   *The Implication:* This is a much harder optimization problem. If the generated workflow contains a single logic error, syntax flaw, or incorrect operator definition, it results in a $0\%$ accuracy across the entire test set.
    *   *The Benefit:* This approach minimizes inference latency. For a task with $N$ incoming queries, MetaFlow requires only **1** generation step for the workflow, followed by deterministic execution.

*   **Baselines (ScoreFlow, FlowReasoner) are Instance-Level Solvers:** These methods generate a unique, bespoke workflow $\mathsf{W_p}$ for each individual problem instance $\mathsf{p}$.
    *   *The Implication:* If the model fails to generate a valid workflow for one problem, it only affects that single data point, allowing for higher average accuracy.
    *   *The Cost:* This approach is computationally expensive and scales linearly with dataset size. For $N$ validation instances, these methods require **$N$** full inference generation calls.

*   **The Trade-off:** We argue that the performance gap (approx. 3.7%) is an acceptable trade-off for the orders-of-magnitude reduction in inference cost (1 call vs. $N$ calls) and the portability of the learned meta-strategy. We have updated the text to position MetaFlow not as a direct competitor in "per-instance accuracy" but as a superior solution for "inference-efficient, scalable workflow deployment."

---

> ### Author Response · Authors · 2025-12-04
> **Global Response (Part 2)**
>
> ### 2. Substantiating OOD Generalization with Novel Operator Experiments
> **Addressed concerns from:**
> *   **Reviewer yP2t (W4):** Claimed "OOD generalization" was overstated as MATH and GSM8K share characteristics.
> *   **Reviewer 2R4k (W3, Q1):** Stated that using a new operator is not enough evidence; requested proof of "structural generalization" rather than just conditional text generation.
> *   **Reviewer qGS7 (W1-2, W1-3, Q1-3):** Argued that MATH is not truly OOD and that the base model (Qwen3-8B) might already possess these capabilities without our training.
> *   **Reviewer oAqH (W2-3, W2-4):** Criticized the reliance on case studies and requested systematic statistical analysis of operator generalization.
>
> **Our Response and Revision:**
> We acknowledge this concern that MATH is not truly OOD. To rigorously prove that MetaFlow learns a "meta-policy" for tool composition rather than just memorizing math patterns, we introduced a completely new experimental setup in **$\texttt{Section 5.3 (line 433)}$** centered on the `VectorSearch` operator.
>
> *   **Experiment Design:** We introduced an operator $\texttt{VectorSearch}$, which performs retrieval-augmented generation (RAG).
>     *   *Why this is crucial:* Unlike the training operators ($\texttt{Generate}$, $\texttt{Summarize}$, $\texttt{Revise}$), which are pure text-processing tasks, `VectorSearch` requires **tool calling** and interacting with an external database. This is structurally and functionally distinct from the training datasets,  makes it a fully OOD task, addressing the concern that MATH was "too similar" to GSM8K.
> *   **Statistical Evidence (vs. Case Studies):** Instead of a single example, we performed a large-scale quantitative evaluation. We conducted $100$ independent rollouts using the operator set $\mathsf{Ops} = \{\texttt{Generate}, \texttt{Summarize}, \texttt{Revise}, \texttt{Ensemble}, \texttt{VectorSearch}\}$.
> *   **Results & Analysis:** We evaluated these 100 generated workflows on both a validation set (100 instances) and a test set (100 instances).
>     *   *New Figure 4:* We plotted the sorted validation accuracy against test accuracy. The results show a clear, monotonic improvement, starting from near-zero (where the model failed to use the tool correctly) to surpassing baseline performance as the rollout explored effective compositions.
>     *   *Conclusion:* This quantitatively proves that the model can adapt to a strictly unseen, structurally different operator (Tool Use) simply by reading its interface definition in the prompt, confirming the "Zero-Shot Novel Operator Integration" claim.

---

> ### Author Response · Authors · 2025-12-04
> **Global Response (Part 3)**
>
> ### 3. Ablation Studies, Baselines, and Training Dynamics
> **Addressed concerns from:**
> *   **Reviewer qGS7 (W1-1, Q1-1):** Demanded zero-shot/few-shot baselines for the untrained base model (Qwen3-8B) to isolate the gain from meta-learning.
> *   **Reviewer qGS7 (W2, Q1-2):** Requested separation of SFT vs. RL contributions and sensitivity analysis of training steps.
> *   **Reviewer qGS7 (W4-1):** Noted inconsistency between text (137 steps) and figures (100 steps).
> *   **Reviewer oAqH (W2-2):** Highlighted missing ablation studies.
>
> **Our Response and Revision:**
> We have significantly expanded **$\texttt{Section 5.1}$** and **$\texttt{Section 4.2.1}$** to address the source of our performance gains.
>
> *   **Base Model Baseline Infeasibility (Response to qGS7 W1-1):** We clarified in **$\texttt{Section 4.2.1 (line 293)}$** that a direct zero-shot baseline of the base model `Qwen3-8B` is technically infeasible for this pipeline. The MetaFlow system requires the generation of a complex, syntactically strict  Python-like workflow template.
>     *   *Observation:* Without the SFT "cold start," the base model fails to adhere to the required syntax or correctly invoke the operator API format. Consequently, the execution engine crashes, yielding a $0\%$ success rate. The SFT stage is a prerequisite for *executability*, not just performance optimization.
> *   **Isolating SFT vs. RL (Response to qGS7 W2):** We updated **Figure 3 (Right)** to explicitly map the training dynamics.
>     *   The starting point of the curve ($0.66$ reward) represents the performance of the model **after SFT but before RL**.
>     *   The subsequent rise to convergence demonstrates the specific added value of the Online RL (GRPO) stage, effectively quantifying the incremental contribution of the reinforcement learning phase.
> *   **Clarification of Training Steps (Response to qGS7 W4-1):** We corrected the confusion regarding step counts. We explicitly state in **$\texttt{Section 5.1}$** that while the total training duration was set to $137$ steps, we selected the checkpoint at step $100$ for final evaluation based on validation set performance (early stopping to prevent overfitting). The figures have been updated to reflect this timeline accurately.

---

> ### Author Response · Authors · 2025-12-04
> **Global Response (Part 4)**
>
> ### 4. Implementation Transparency, Reproducibility, and Costs
> **Addressed concerns from:**
> *   **Reviewer yP2t (W3):** Criticized the lack of analysis on and configuration details.
> *   **Reviewer qGS7 (W4-2, Q2):** Called the SFT data "opaque" (how was it generated? what prompts?) and demanded cost/latency reports (N, k, tokens).
> *   **Reviewer oAqH (W3):** Noted missing learning rates, batch sizes, and hardware details.
>
> **Our Response and Revision:**
> We have conducted a massive overhaul of the Appendices to ensure the paper meets the standards of reproducibility.
>
> *   **SFT Data Transparency (Appendix B.1):** We added **$\texttt{Appendix B.1 (line 937)}$** to describe the SFT dataset construction in granular detail.
>     *   We explain the use of `Qwen-Max` as the teacher model.
>     *   We provide the exact prompts used to synthesize the 1,300 workflow samples.
>     *   We detail the verification process used to filter out invalid workflows before training.
> *   **Hyperparameters & Configuration (Appendices A, B, C):** We added:
>     *   **$\texttt{Appendix A.1 (line 671)}$:** Full definition of all operators and their code interfaces.
>     *   **$\texttt{Appendix A.2 (line 876)}$:** Exact Input/Output JSON formats for the pipeline.
>     *   **$\texttt{Appendix B.2 (line 1119)}$:** Exact LoRA configurations (rank, alpha, target modules).
>     *   **$\texttt{Appendix C (line 1134)}$:** Online RL environment settings, including the rollout parameter ($k=8$) and step batch size ($24$).

---

> ### Author Response · Authors · 2025-12-04
> **Global Response (Part 5)**
>
> ### 5. Presentation Quality
> **Addressed concerns from:**
> *   **Reviewer yP2t (W2):** Figures lacked visual details; Figure 5 was unnecessary.
> *   **Reviewer qGS7 (W5):** General figure quality issues.
> *   **Reviewer oAqH (W3):** Code listings lacked highlighting; Figure 1 was low quality. **(Response to our rebuttal):** The bolded items in Table 1 must be replaced.
>
> **Our Response and Revision:**
> We have professionally redesigned the visual components of the paper and updated the baselines.
>
> *   **Inclusion of FlowReasoner:** We initially omitted it because it follows the same **instance-level** approach as ScoreFlow and primarily focuses on coding tasks.  We have added **FlowReasoner** to our experimental comparison in **$\texttt{Table 1}$** and the related discussion in **$\texttt{Section 5.4}$**. We analyze it alongside ScoreFlow as another instance-level approach, confirming that while it is accurate, it suffers from the same efficiency bottlenecks we aim to solve.
> *   **Figure 1 (Revised):** We replaced the text-heavy diagram with a high-quality vector graphic in **$\texttt{Section 1 (line 57)}$**. It now visually contrasts the architectures of "Prompting," "Instance-level Search (ScoreFlow/FlowReasoner)," and "Cross-task Generation (MetaFlow)."
> *   **Figure 2 (Revised):** We created a new schematic in **$\texttt{Section 4.1 (line 248)}$** that clearly illustrates the basic operators and the data flow through the MetaFlow pipeline.
>    **Appendix Formatting:** We applied syntax highlighting to all code listings in the appendix to improve readability.
>  **Table1 Formatting:** The bold formatting has been removed, and we now use a subtle background color to distinguish MetaFlow's results.
>
> ### 6.  **Detailed Justification of Cost & Search Strategy (Response to qGS7 W4-3, W4-4):**
>
> We have added a comprehensive analysis in **$\texttt{Section 5.3 (line 433)}$** to defend the efficiency and scalability of our search strategy, leveraging the new `VectorSearch` experiments as evidence.
>
> +  **Parallel Efficiency vs. Iterative Search:** We explicitly contrast our method with iterative baselines like AFlow. While AFlow relies on serial improvements (generate $\rightarrow$ evaluate $\rightarrow$ revise), which accumulates high latency, our "best-of-20" strategy is **fully parallelizable**. The generation of candidate workflows and their execution on validation instances can occur simultaneously.
> +   **Evidence from VectorSearch:** In the new experiments involving the `VectorSearch` operator, we expanded the search space to $100$ rollouts. As visualized in **Figure 4**, we observed that both validation and test accuracy improve monotonically as the rollout size increases.
> +  **Cost Amortization:** We emphasize that this search cost is a **one-time investment per task**. Once the optimal workflow $\mathsf{W^*_C}$ is identified from the validation search, it is frozen. For all subsequent incoming queries, the cost is strictly **single-inference**. This makes our approach significantly more scalable for deployment compared to instance-level methods (like ScoreFlow), which must pay the search cost for *every single new user query*.

---

### Meta-Review · Area_Chair_ks7P · 2026-01-07

**Summary:**

This paper proposes MetaFlow, which learns to generate a single reusable workflow per task category (cross-task workflow generation) via SFT + RLVR (GRPO), aiming to avoid per-instance re-optimization at inference. Reviewers agreed the problem is timely and the framing is interesting, but initial reviews criticized weaker accuracy than strong baselines (e.g., ScoreFlow/AFlow), unclear “meta-learning/OOD” claims, missing baselines/ablations, and poor reporting of costs and implementation details. The rebuttal substantially improves the paper: it clarifies the cross-task vs instance-level paradigm (and the associated accuracy/efficiency trade-off), adds a more convincing novel-operator evaluation (VectorSearch/tool-calling), adds missing baselines/ablations discussion (SFT as executability prerequisite; RL contribution via training curves), and overhauls reproducibility and presentation. One reviewer explicitly increased their score after the revision.

**Reviewer Concerns:**

Addressed by rebuttal:
- Definition/positioning: Clearer articulation that MetaFlow is a cross-task generator (one workflow reused across instances), not an instance-level solver; this explains the accuracy vs robustness trade-off.
- OOD/operator generalization evidence: Added a more genuinely OOD operator setup (VectorSearch with tool use) with quantitative rollouts rather than only case studies.
- Reproducibility and missing details: Added SFT data generation prompts, filtering, hyperparameters, LoRA/RL settings, and clarified the training-step inconsistency.
- Presentation quality: Figures and table formatting issues were fixed; FlowReasoner added as a baseline; code listings improved.
- Some reviewer sentiment improved: At least one reviewer (oAqH) raised their score to 4 after the revisions.

Still outstanding / not fully convincing:
- Accuracy gap remains: Even with the “different paradigm” explanation, MetaFlow still underperforms instance-level baselines on reported averages; the paper needs to be careful not to oversell “competitive/SOTA.”
- Baseline feasibility claims: The argument that the base model baseline is “infeasible” because workflows don’t execute without SFT is plausible, but it weakens causal attribution. It would be stronger with at least a controlled comparison (e.g., SFT-only vs SFT+RL with the same executability constraints, plus explicit execute-rate reporting).
- Cost accounting is still a bit fuzzy: The paper now explains parallelizable best-of-N search and amortization, but the actual end-to-end cost (rollouts, validation set size, operator calls) still sounds hard to compare cleanly against baselines.

**Reviewer Scores:**

If reviewers read the rebuttal:
- yP2t: 2 → 2–3 (some concerns addressed, but likely still negative due to accuracy gap / novelty skepticism).
- 2R4k: 4 → 4 (presentation/baselines improved; still cautious).
- qGS7: 2 → 2–3 (some major issues addressed, but core attribution/cost questions likely remain).
- oAqH: 2 → 4 (explicitly raised after rebuttal).

---

### Decision · Program_Chairs · 2026-01-26

Reject